# Fusion surface structure, function, and dynamics of gamete fusogen HAP2

Juan Feng[1,2†], Xianchi Dong[1,2†], Jennifer Pinello[3†], Jun Zhang[3†], Chafen Lu[1,2], Roxana E Iacob[4], John R Engen[4], William J Snell[3*], Timothy A Springer[1,2*]

[1]Department of Biological Chemistry and Molecular Pharmacology, Harvard Medical School, Boston, United States; [2]Program in Cellular and Molecular Medicine, Children's Hospital Boston, Boston, United States; [3]Department of Cell Biology and Molecular Genetics, University of Maryland, College Park, United States; [4]Department of Chemistry and Chemical Biology, Northeastern University, Boston, United States

**Abstract** HAP2 is a class II gamete fusogen in many eukaryotic kingdoms. A crystal structure of *Chlamydomonas* HAP2 shows a trimeric fusion state. Domains D1, D2.1 and D2.2 line the 3-fold axis; D3 and a stem pack against the outer surface. Surprisingly, hydrogen-deuterium exchange shows that surfaces of D1, D2.2 and D3 closest to the 3-fold axis are more dynamic than exposed surfaces. Three fusion helices in the fusion loops of each monomer expose hydrophobic residues at the trimer apex that are splayed from the 3-fold axis, leaving a solvent-filled cavity between the fusion loops in each monomer. At the base of the two fusion loops, Arg185 docks in a carbonyl cage. Comparisons to other structures, dynamics, and the greater effect on *Chlamydomonas* gamete fusion of mutation of axis-proximal than axis-distal fusion helices suggest that the apical portion of each monomer could tilt toward the 3-fold axis with merger of the fusion helices into a common fusion surface.

DOI: https://doi.org/10.7554/eLife.39772.001

*For correspondence:
wsnell1@umd.edu (WJS);
springer_lab@crystal.harvard.edu
(TAS)

†These authors contributed equally to this work

Competing interests: The authors declare that no competing interests exist.

## Introduction

Sexual reproduction in eukaryotes is key to evolution of organismal complexity and diversity. A defining event of sex is fusion of the plasma membranes of two haploid gametes to generate a new diploid cell, yet we still know little about the molecular mechanism of gamete fusion. The single-pass transmembrane protein HAP2/GCS1 is essential for gamete fusion in organisms across eukaryotic kingdoms, and was likely the ancient sexual fusogen used by primitive eukaryotes before the eukaryotic radiation (*Wong and Johnson, 2010*). HAP2 is also important as a target of vaccines that block the obligate sexual life cycles of parasites including *Plasmodium*, and hence block transmission of malaria (*Angrisano et al., 2017*).

Recent work on HAP2 from the green alga *Chlamydomonas*, the flowering plant *Arabidopsis*, and the ciliate *Tetrahymena* shows that HAP2 is, in common with fusogens used by certain enveloped viruses to enter host cells, a class II fusion glycoprotein (*Fédry et al., 2017*; *Pinello et al., 2017*; *Valansi et al., 2017*). Class II fusogens are characterized by their mainly β-sheet-containing-domains I, II, and III (referred to here as D1, D2, and D3) (*Harrison, 2015*; *Kielian and Rey, 2006*). A stem region connects these globular domains to two transmembrane domains in flaviviruses (*Zhang et al., 2013*) or a single-pass transmembrane domain and a cytoplasmic domain in eukaryotes. Although sequence homology is not detectable among class II fusogens from different virus families or with HAP2, similar three-dimensional structures and functions suggest that they diverged from a common ancestor (*Fédry et al., 2017*; *Pinello et al., 2017*; *Valansi et al., 2017*). As with viral class II proteins,

HAP2 functions uni-directionally. Thus, in *Chlamydomonas*, HAP2 is expressed and required for fusion only in *minus* mating type gametes, and not in *plus* mating type gametes.

In the classic model of viral class II fusogen function, glycoproteins are tightly packed in a regular icosahedral lattice overlying the viral lipid bilayer in which their transmembrane domains are embedded (*Harrison, 2015*; *Kielian and Rey, 2006*). After the virus adheres to host cell receptors and is internalized, the low pH of the endosome triggers reconfiguration of the fusogen from its pre-fusion state into an activated, monomeric state. In this state, a hydrophobic 'fusion loop' is exposed at the apical tip of D2, which is the most distal domain from the viral membrane. The tip of the exposed fusion loop then anchors to the target endosomal membrane, creating a protein bridge between the viral and host-cell membranes. This in turn initiates a concerted series of steps leading to trimerization. In a first step, monomers associate in parallel along a trimer axis lined by D1 and D2 (*Liao et al., 2010*). Then, in a change of D1-D3 orientation, D3 folds back onto D1 and the lower part of D2 and a portion of the stem zippers along the upper part of D2. In the final steps of conformational change, which are yet to be resolved structurally, the remainder of the stem containing hydrophobic elements packs against the trimer and the target membrane, and drags the transmembrane domain anchored to the viral membrane into intimate contact with the fusion loops moored in the target endosomal membrane. Close approach of the viral and host lipid bilayers and their ensuing distortions during the final stages of stem zippering and hydrophobic element approach are thought to destabilize the membranes enough to fuse into a pore that connects the viral contents with the cytoplasm of the host cell. All trimeric class II structures are commonly termed 'post-fusion'; however, since some of them may represent intermediates that are on the pathway to fusion, we use the term 'fusion state' structures here.

Comparisons of pre-fusion and fusion-state structures of viral class II fusion proteins show not only a fold-back of D3 onto D1 and some features of stem zippering, but also a change in D1-D2 orientation (*Harrison, 2015*; *Kielian and Rey, 2006*). However, following trimerization, little is known about changes in the orientation of fusion loop-bearing D2 in the pathway towards the final fusion conformation. D2 domains in trimers from some viral class II fusogens pack tightly against their neighbors at the 3-fold axis, while others splay widely or to intermediate extents from the 3-fold axis (*Harrison, 2015*; *Luca et al., 2013*). Closely packed fusion loops could provide a larger, unified fusion surface at the tip of the trimer necessary for strong anchoring in the target lipid bilayer. Because only a small number of the residues in the fusion loop of *Chlamydomonas* HAP2 were previously resolved (*Fédry et al., 2017*), we lack important information about the location and structural relationships of putative fusion loop residues and the degree of HAP2 D2 splaying in the trimer.

Here, we report a 2.6 Å trimeric, fusion-state crystal structure of *Chlamydomonas* HAP2 in which the fusion loops are completely resolved. To our surprise, and unlike viral fusogens, hydrophobic fusion loop residues in HAP2 are exposed on three separate helices in each monomer - - α1, η1, and α2 - - positioned to interact with the target cell (*plus* gamete) membrane. Mutational analyses show that the α1 and α2 helices, which are parallel to one another, have a much more critical function in fusion than the η1 helix, which is tangential to and more distant from the central axis. These results, the large amount of splaying of the fusion loop of each protomer from the trimer central axis, and hydrogen-deuterium exchange (HDX) results showing flexibility within D2 suggest that the crystallized form of HAP2 is an intermediate in the structural transition pathway for membrane fusion. We propose that during the final stages of fusion, the apically localized fusion loops of HAP2 tilt towards the trimer core to form a more compact fusion surface.

## Results

### Overall HAP2 trimer conformation

*Chlamydomonas* HAP2 glycoprotein (ectodomain residues 23 – 582) with a C-terminal His tag was secreted from *Drosophila* S2 cells, purified by Ni-affinity chromatography and gel filtration, and crystallized. The structure was refined to 2.6 Å with $R_{work}$ and $R_{free}$ of 23.2% and 28.1%, respectively (*Supplementary file 1*). Three monomers in the asymmetric unit form a trimer and pack against one another through D1 and D2 at the 3-fold axis (*Figure 1A*). In further similarity to other class II fusion-state structures, a U-turn between D1 and D3 enables D3 to pack against the outer faces of D1 and D2. D2 narrows at a waist that divides two subdomains we term D2.1 and D2.2. The subdomains

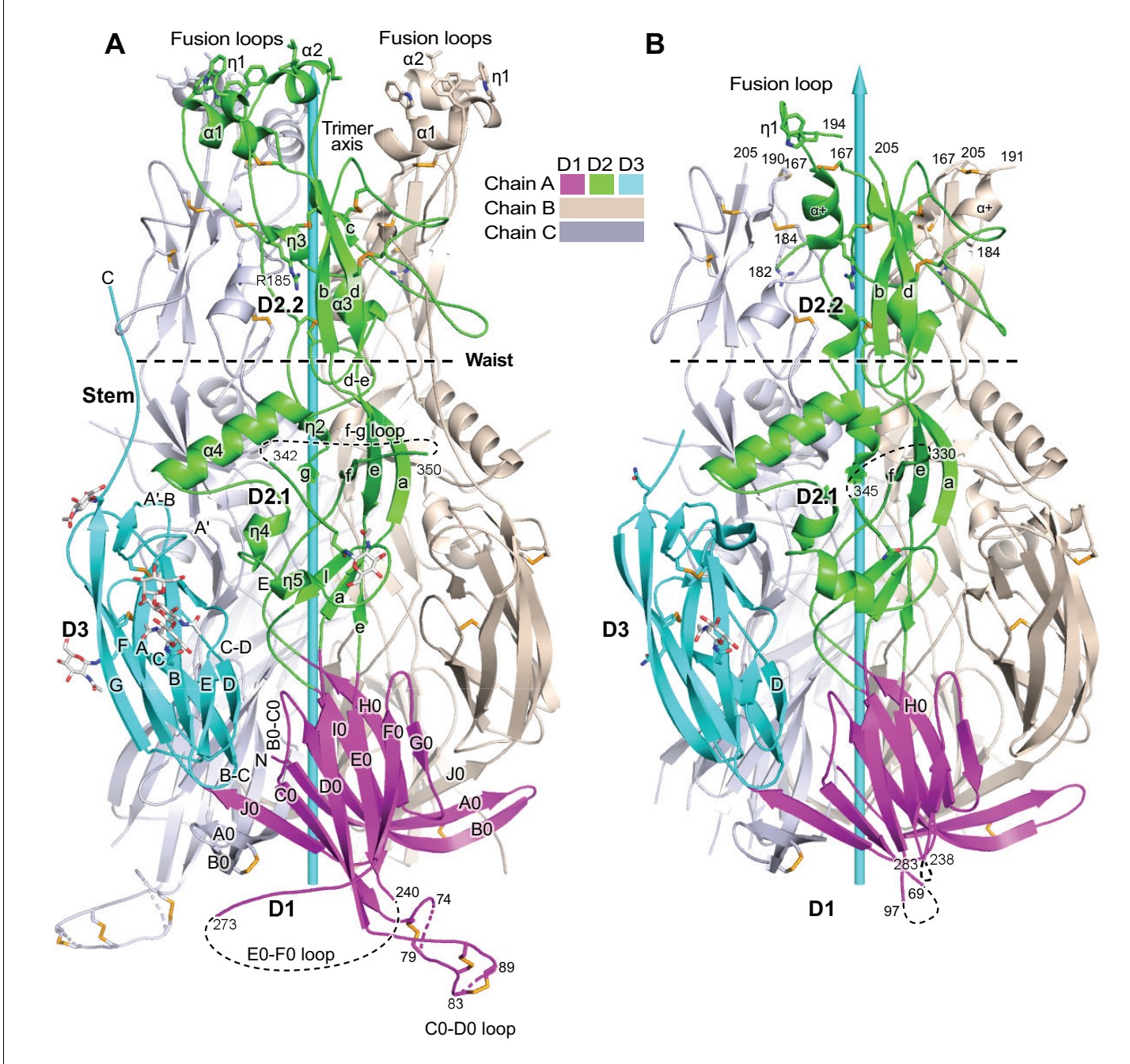

**Figure 1.** Overall structure of HAP2. (A, B) Ribbon diagrams of current (A) and previous (B) trimer structures. Apically exposed residues in the fusion loops, disulfide bonds, and Asn-linked glycans are shown in stick with orange sulfurs, red oxygens, blue nitrogens, and white carbons in glycans. Loops have been smoothed for simplicity. The trimer 3-fold axis is shown with a cyan arrow. Regions of missing electron density, except for the fusion loops in (B) are shown as dashed lines. Residues adjacent to gaps are numbered.

DOI: https://doi.org/10.7554/eLife.39772.002

contain separate β-sheets and α-helices. D3 packs against D2.1, which locates midway along the 3-fold axis in trimers. D2.2 at the 'apical', narrower end of trimers bears three fusion helices, α1, η1, and α2, in fusion loops 1 and 2 between β-strands c and d. Following D3, a stem segment binds in a groove in D2 of a neighboring monomer. Although the stem is formed in part by five ordered residues of the C-terminal His tag, this region of HAP2 is poorly conserved in sequence and the stem extends in the expected direction towards the fusion loops. In full-length HAP2, the stem includes

48 additional residues including a hydrophobic region and is followed by the transmembrane domain.

Domains D1, D2.1, and D3 in our HAP2 structure (*Figure 1A*) orient almost identically as previously described (*Fédry et al., 2017*) (*Figure 1B*), as may be seen by comparing the secondary structure elements marked in *Figure 1B* to those in *Figure 1A*. In contrast, D2.2 orientation differs, as seen by comparing the separation of β-strands b and d from the 3-fold axis (*Figure 1A,B*). Compared to the 3.3 Å structure, the 2.6 Å structure shows differences in sidechain orientation, peptide backbone orientation, hydrogen bonds, and sequence-to-structure register (Materials and methods). Either as a consequence of omission of protease treatment here, differences in the crystal lattice, or higher resolution, we can visualize 50 more residues per monomer. Importantly, we can completely trace the fusion loops, which form three helices with apically-pointing hydrophobic residues in each monomer (*Figure 1A*). Of nine total fusion helices, only one was resolved in the previous structure, and it differed in orientation (*Figure 1B*), accounting for the markedly different appearance of the two structures in the apical portion of D2.2 (*Figure 1A,B*). Additionally, other loops are resolved better in our structure, including at the base of trimers (*Figure 1A,B*). Long loops are prominent in HAP2, differ in position among HAP2 from different species, and contrast with the shorter loops found in viral fusion proteins.

## Monomer and trimer dynamics

To obtain complementary structural information, we measured HDX for HAP2 in both its monomeric and trimeric forms. Purified, monomeric HAP2 fortuitously trimerized into a fusion state during crystallization. To obtain a trimeric, fusion-state form for HDX, we incubated monomeric HAP2 with dodecylmaltoside detergent to bind to its fusion loops and simulate membrane engagement. Multi-angle light scattering showed that monomeric HAP2 contains 6% glycan, and that the glycoprotein mass of the main peak after detergent treatment was 2.99-fold greater than the monomer, in excellent agreement with trimer formation (*Figure 2—figure supplement 1*).

HDX measures exchange of protein backbone NH hydrogens with deuterium in deuterated water ($D_2O$). The main factors that slow exchange are burial from solvent and hydrogen bonding, primarily to backbone NH groups (*Bai et al., 1993*). Because exchange measures solvent exposure and lack of backbone hydrogen bonds, it approximates flexibility. HDX kinetics were measured over time periods from 10 s to 4 hr. After quenching and pepsin digestion, deuterium incorporation was measured in individual peptides by mass spectrometry (*Figure 2A–C* and *Figure 2—figure supplement 2 and 5*). Trimer HDX at 1 min is displayed on the ribbon cartoon of one monomer with nearby regions of other monomers shown as transparent surfaces (*Figure 2A–C*). As discussed later in results, the most dynamic regions of D1, D2.2, and D3 surprisingly lie more on their faces directed toward, rather than away from, the 3-fold axis. The apical portion of D2.2 containing its fusion loops was among the most rapidly exchanging regions in HAP2. The small β-sheets in D2.1 and D2.2 surprisingly exchanged more slowly than the larger β-sheets in D1 and D3. The waist between D2.1 and D2.2 is thin and contained more rapidly exchanging regions. Comparison between monomer and trimer HDX revealed no overall effect of trimerization; instead, differences were limited to specific regions (*Figure 2D*).

## Structure and dynamics of domains 1 and 3

We now describe HDX results and 3D structure together, domain by domain. D1 forms the base of the HAP2 trimer. Its first two β-strands, A0 and B0, form a long disulfide-bonded β-ribbon that is unusual in extending to the β-sheet on the other face of D1 in a neighboring monomer (*Fédry et al., 2017*) (*Figure 1A*). In agreement, HDX is lower in this region in the trimer relative to the monomer (*Figure 2D*).

Two long, partially disordered loops decorate D1 on the outer perimeter of the trimer base (*Figure 1A*). Six disulfide-bonded cysteines in the C0-D0 loop and 18 residues in the E0-F0 loop are newly built in our structure (*Figure 1A,B*). Both loops extend away from the trimer axis and occupy a large solvent cavity in crystals. The E0-F0 loop has a Ser, Thr, and Pro-rich sequence that is predicted to be O-glycosylated and four predicted N-glycosylation sites in the disordered region (*Figure 3*).

Domain 3 has two β-sheets that sandwich together with a hydrophobic core. D3 is termed immunoglobulin-like; however, it should be classified as an FN3 domain, since its D β-strand joins the GFC

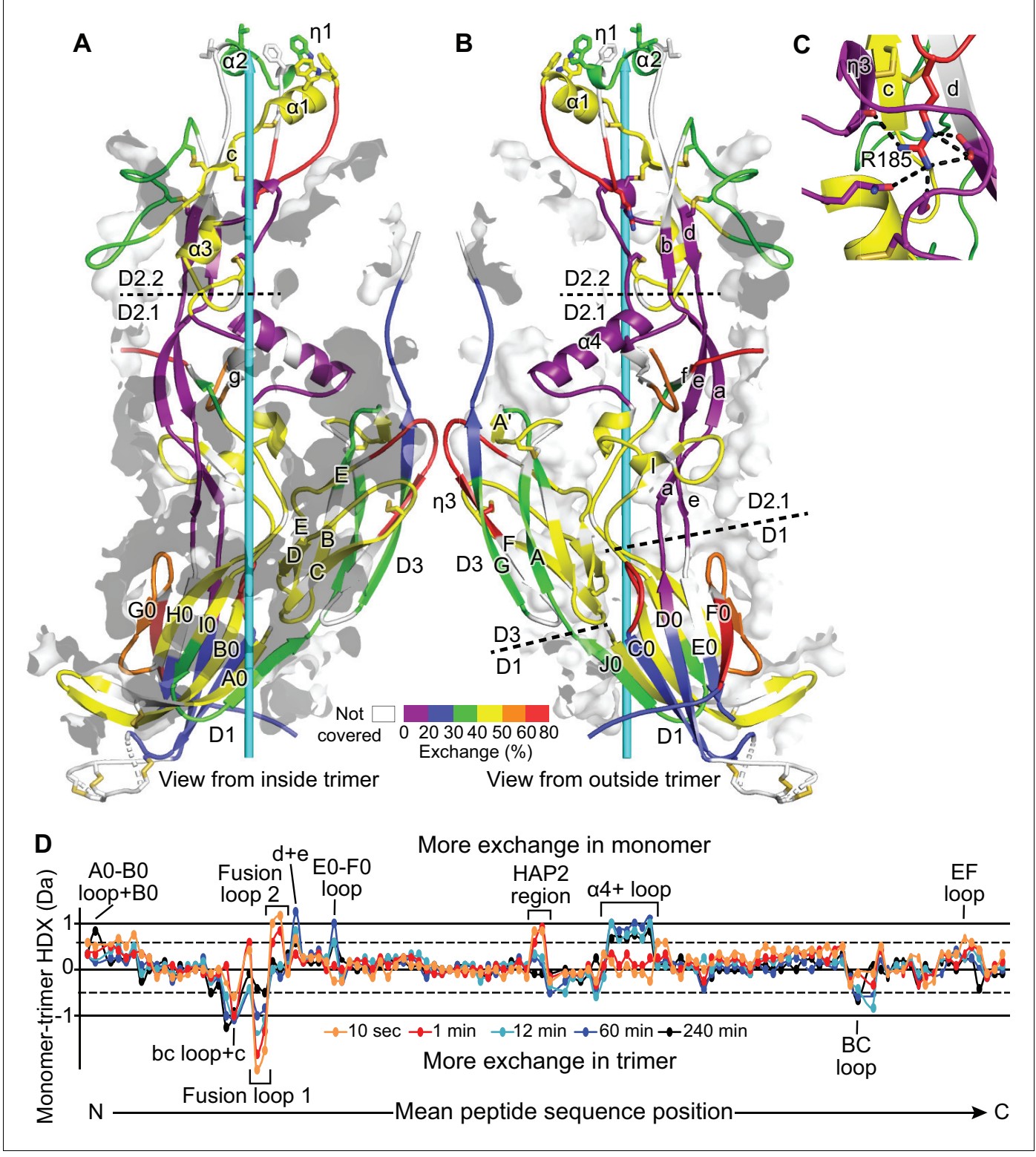

**Figure 2.** HDX. (**A,B**) One monomer of the trimer is shown viewed from inside (**A**) and outside (**B**) the 3-fold axis (cyan arrow) with HDX at 1 min. color-coded on the ribbon diagram. Secondary structure elements within each domain are divided and labeled according to whether they are closer to the 3-fold trimer axis (**A**) or further from the axis (**B**). Transparent surfaces of other monomers within 4 Å appear white and grey on their exteriors and interiors, respectively. (**C**) Detail of the Arg185 carbonyl cage at 1 min of HDX. Hydrogen bonds are dashed. (**D**) Difference in trimer and monomer HDX for every shared peptide is shown in colored curves (*Houde et al., 2011*). Key structural positions are marked.

*Figure 2 continued on next page*

*Figure 2 continued*

DOI: https://doi.org/10.7554/eLife.39772.003

The following figure supplements are available for figure 2:

**Figure supplement 1.** Purification and Trimerization of HAP2.

DOI: https://doi.org/10.7554/eLife.39772.004

**Figure supplement 2.** Trimer and monomer HDX and secondary structure of HAP2.

DOI: https://doi.org/10.7554/eLife.39772.005

**Figure supplement 3.** Deuterium incorporation plots for all 120 peptic peptides that were compared between the monomer (red) and trimer (blue).

DOI: https://doi.org/10.7554/eLife.39772.006

**Figure supplement 4.** See legend for *Figure 2—figure supplement 3* .

DOI: https://doi.org/10.7554/eLife.39772.007

**Figure supplement 5.** See legend for *Figure 2—figure supplement 3* .

DOI: https://doi.org/10.7554/eLife.39772.008

β-sheet as in FN3 domains rather than the ABE β-sheet as in Ig domains (*Figure 1A*). Search for structurally homologous domains (*Holm et al., 2008*) shows that D3 closely matches the elongated FN3 domains found in complement components C3, C4, and C5, which are termed macroglobulin domains.

HDX shows signatures of the conformational change from the monomeric to the trimeric state in which D3 associates with D1 and D2. Faster exchange in trimers in the B-C loop (*Figure 2D*) which is well exposed to solvent but is at the end of D3 facing D1 (*Figure 1A*) suggests a structural alteration in monomers, consistent with their hypothesized more linear D1-D3 interface.

## Structure and dynamics of domain 2

D2 is formed by both β-sheets and α-helices. Of its two subdomains, larger D2.1 has two β-sheets that extend one another and pack against α-helices and long loops. Smaller D2.2 extends ~1/3 the distance of the trimer axis like D2.1, but is thinner and has a single β-sheet that is stabilized by interactions with loops and short α-helices. The slowest exchanging regions in HAP2 are found in D2.1 near its junction with D2.2, where D2.1 has a substantial hydrophobic core underlying the right-angle junction between the slow-exchanging α4-helix and aef β-sheet (*Figure 1A* and *Figure 2B*). Slowly exchanging regions found in the thin D2.2 subdomain include the lower portions of β-strands b and d and a loop with a η3-helix. Together, these elements in D2.2 form a carbonyl cage that docks Arg185, which is present in a rapidly exchanging loop (*Figure 2B,C*). The apex of D2.2 including the entire β-strand c, fusion loops 1 and 2, and the portions closest to the 3-fold axis are in rapid to moderate HDX, suggesting flexibility (*Figure 2A,B*). In agreement with slower exchange in trimeric than monomeric HAP2, the d and e β-strands and the α4-helix and following loop (*Figure 2D*) are buried in the trimeric assembly including in an interface with D3 (*Figure 1A*).

We built ordered portions of the long f-g loop in D2 that protrudes outward from the trimer midway along its 3-fold axis (*Figure 1A*). HDX showed that the portion that could not be built is disordered (*Figure 2—figure supplement 2 and A*). Based on its sequence, *Plasmodium* HAP2 contains an even longer f-g loop with four cysteines and a long loop between η5 and the l β-strand in D2 (*Figure 3*).

The highly conserved 'HAP2-GCS1 domain' (PFAM PF10699), residues 352–399, lies close to the 3-fold trimer axis at the junction between D2.1 and D2.2, and largely contains loops and short η and α-helices (*Figure 4A*). Highly conserved sidechains in this region form hydrophobic cores that underlie the α4-helix and aef β-sheet in D2.1 and the carbonyl cage in D2.2 (*Figure 4B*). Conserved polar residues form many hydrogen bonds including Gln379 that hydrogen bonds to Arg185 in the carbonyl cage. Nonetheless, the majority of HAP2-GCS1 residues in D2.2 (360 – 380) show >40% HDX at 1 min. Thus, this region has exposed backbone amides, suggesting it has access to conformational change. Malleability of the 360 – 380 region is further suggested by lack of effect on HAP2 function of mutations of conserved residues Asp367 and Lys368 (*Liu et al., 2015*).

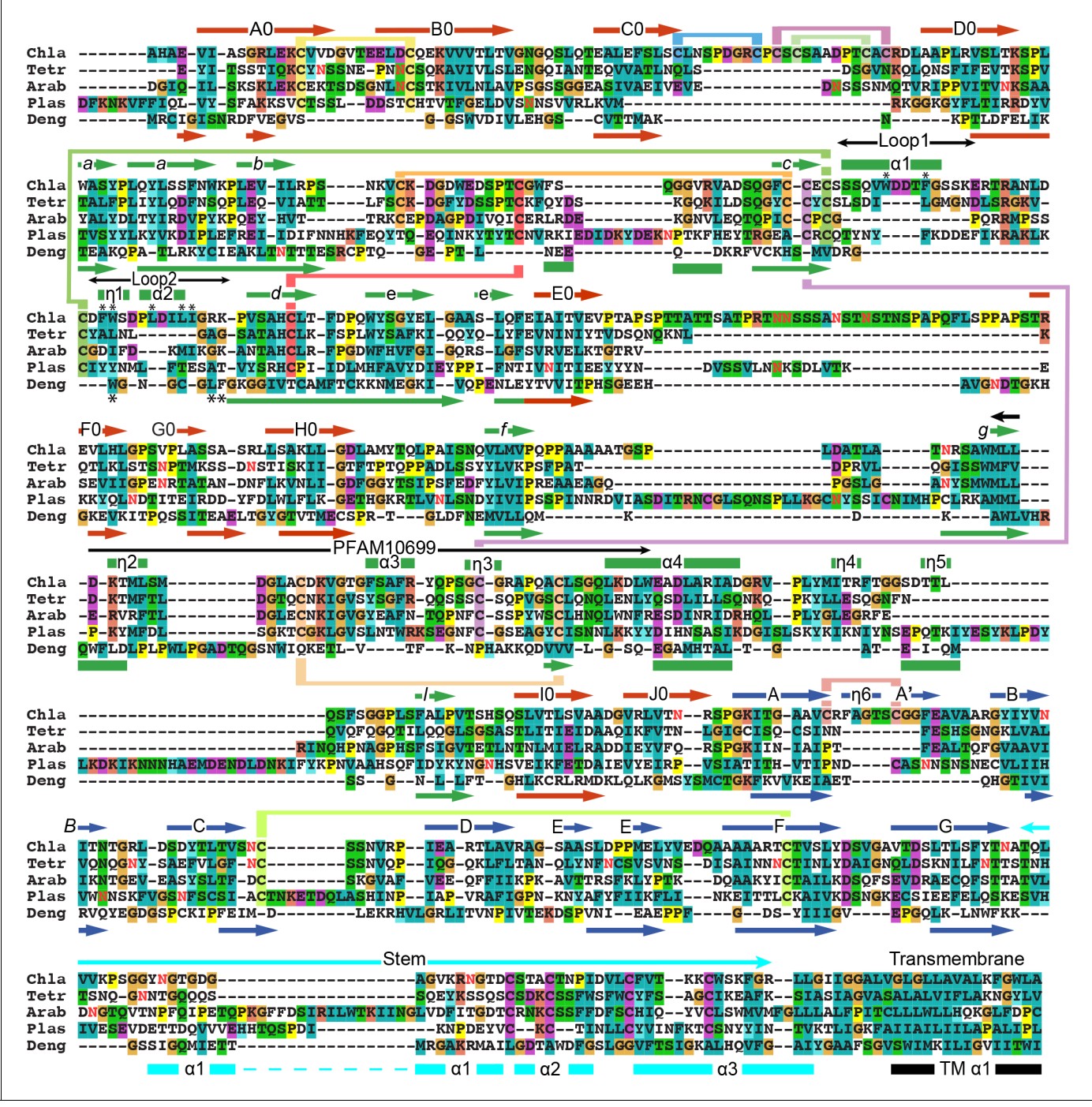

**Figure 3.** Sequence and structure-based alignments of HAP2 from diverse species and *Dengue 2* class II viral fusogen. HAP2 sequences were aligned with MAFFT (**Katoh and Standley, 2013**) with G-INS-I, BLOSUM30, 2.5 opening penalty, and 0.12 offset value. HAP2 and *Dengue 2* fusion protein (PDB ID 1OK8) domains 1 + 2 and domain 3 were separately aligned structurally using RaptorX (**Wang et al., 2013**) to obtain structurally aligned sequences. The two alignments, each containing HAP2, were combined with MAFFT, and closed up manually in loop regions. Missing stem and transmembrane domains were aligned by sequence. Secondary structure elements of HAP2 and *Dengue* (**Zhang et al., 2013**; **Modis et al., 2004**) color coded by domain are shown above and below the alignment, respectively. Predicted glycosylated Asn residues are shown in red. Chla, *Chlamydomas reinhardtii*; Tetra, *Tetrahymena thermophila*; Arab, *Arabidopsis thaliana*; Plas, *Plasmodium berghei*; Deng, *Dengue 2*.

DOI: https://doi.org/10.7554/eLife.39772.009

The following figure supplements are available for figure 3:

**Figure supplement 1.** HAP2 sequences from representatives of different phylogenetic groups, selected for short length, and aligned with MAFFT as described in Main Text Fig. 3.

*Figure 3 continued on next page*

*Figure 3 continued*

DOI: https://doi.org/10.7554/eLife.39772.010

**Figure supplement 2.** See legend for *Figure 3—figure supplement 1*.

DOI: https://doi.org/10.7554/eLife.39772.011

## Hydrophobic fusion residues are displayed on three helices

In common with the fusion loops of most viral class II fusogens, the fusion loops in HAP2 lie between the c and d β-strands. The fusion loops in HAP are unusually long at 39 residues and have three helices. Each helix, α1 in loop 1, and η1 and α2 in loop 2, displays a set of two to three hydrophobic residues that project apically (*Figure 4D–H*). Loop 1 extends from Cys167 to Arg185 and contains the α1-helix with Trp173 and Phe177 in an apical orientation. Loop 2 lies between Arg185 and the beginning of β-strand d. It contains the η1-helix with Phe192 and Trp193 that orient apcally, and the α2-helix with Leu197 and Leu200 that orient apically and Ile201 that orients outward. The three helices in each monomer have orientations similar to three sides of a box (lines in *Figure 4D* mark helix axes of the right-most monomer), with axes of α1 and α2 pointing toward the 3-fold axis while the η1-helix axis forms the outer side, more distal from the 3-fold axis. The division between fusion loops is defined by a basally projecting loop with Arg185 at its tip. The Arg185 sidechain guanido group docks through six hydrogen bonds to its carbonyl cage formed by backbone and sidechain carbonyl oxygens in the core of subdomain D2.2 (*Figures 2C* and *4C,G*).

The architecture of the 3-helix fusion surface in each HAP2 monomer is well supported by a hydrophobic core that underlies and knits together the α1, η1, and α2-helices and a robust network of hydrogen bonds (*Figure 4E*). Notable hydrogen bonds include those made by the Ser168 and Asp191 sidechains with backbone NH groups of the α1 and η1 helices, respectively, that stabilize these helices by capping their N-termini. Additionally, Thr-176 caps the η1 helix and also hydrogen bonds to the α1-helix to link these helices.

HAP2 ectodomain monomers appear to utilize their fusion loops to bind phospholipid vesicles (*Fédry et al., 2017*) and detergent (*Figure 2—figure supplement 1*) as shown by induction of trimerization. Thus, in the trimeric HAP2 state, the hydrophobic residues on the fusion helices in each monomer should also be able to interact with the lipid bilayer of the *plus* gamete. In the crystal structure, however, D2.2 splays away from the 3-fold axis so that the 3-helix fusion surfaces of monomers are separated from one another by solvent (*Figure 4A and D*). Because of the large hydrophilic surface exposed on the faces of each monomer surrounding the 3-fold axis, deformation of lipids to fit into this large cavity, or the presence of water in the cavity, would limit the depth to which HAP2 trimers could insert in the hydrophobic core of the membrane bilayer during membrane fusion. These apparent barriers raise the possibility that our crystal structure represents an early fusion state of HAP2 that precedes a more mature state in which reorientation could allow the fusion loops in each monomer to approach and close up at the 3-fold axis to form a common fusion surface.

## Flexibility in D2 in HAP2

Comparisons among HAP2 and viral fusion state crystal structures and among monomers in these structures reveal motions that affect how closely fusion loops approach one another at the 3-fold axis (*Figures 5* and *6*). The two HAP2 crystal structures superimpose extremely well on D1, D2.1, and D3 (*Figure 5B*); however, orientations at their D2.1-D2.2 junctions differ (*Figure 5A*). While the three monomers in our HAP2 structure have essentially identical D2.1-D2.2 orientations, monomers in the previous structure differ; *Figure 5A–B* displays one monomer from our structure and two from the previous structure. We also compare HAP2 to fusogens from three flaviviruses (*Figure 5C–F*). *Figure 5C–D* compares three monomers from one of the two trimers in the *Tick-borne encephalitis* virus structure (*Bressanelli et al., 2004*). *Figure 5E–F* compares monomers from the trimeric *Dengue 1* and *St. Louis encephalitis* virus structures (*Luca et al., 2013*; *Nayak et al., 2009*), which have only one crystallographically unique monomer in their asymmetric units. The most fusion loop-proximal residue that is conserved in position between HAP2 and flaviviral fusogens as shown by superposition of D2.2 (*Wang et al., 2013*) is marked with a Cα sphere and its distance from the 3-fold axis is shown for each monomer in (*Figure 5*). Interestingly, the HAP2 and *Tick-borne encephalitis* fusogens show similar flexibility at the D2.1 and D2.2 junction (*Figure 5A,B*). Their flexibility is

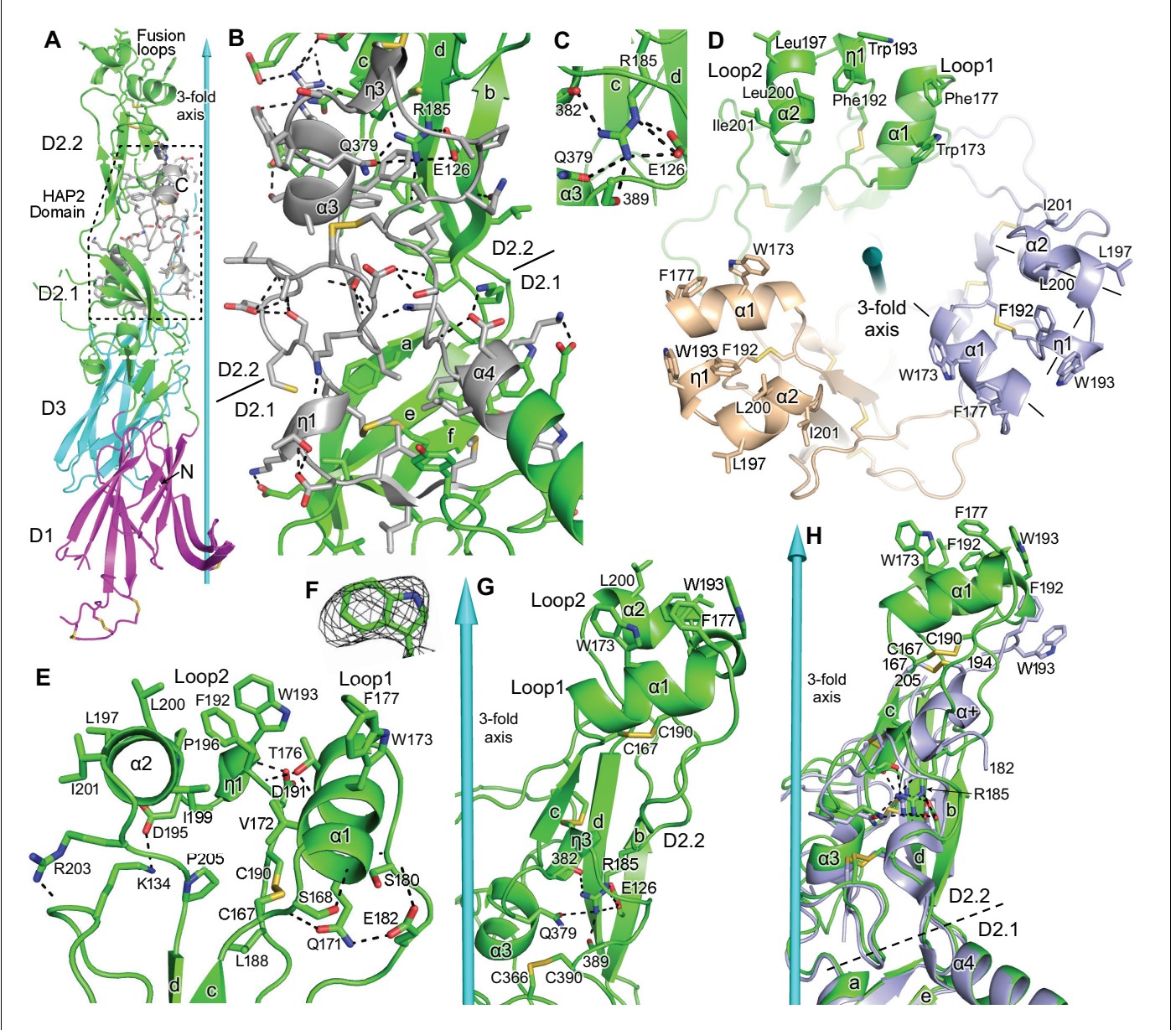

**Figure 4.** Domain D2.2. HAP2 is in ribbon cartoon with selected sidechains and disulfides shown in stick, with nitrogens blue, oxygens red, and sulfurs gold. Cyan cones and cylinders show the trimer 3-fold axis. Hydrogen bonds are shown as dashed lines. (A) One monomer, with the HAP2 domain (PFAM 10699) shown in silver with dashed outline. (B) Detail of PFAM 10699 (silver) in an orientation rotated about 180° from that in (A) All sidechains in PFAM 10699 and their hydrogen bonds, including sidechain-backbone hydrogen bonds, are shown; however, backbone atoms participating in these hydrogen bonds are omitted. Only the most conserved residues in PFAM 10699, and selected interacting residues, are labeled. (C) Detail of Arg185 in its carbonyl cage. (D) Apical view of the fusion loops. Loops are labeled and 3-letter amino acid codes are used in the upper monomer and 1-letter codes are used in lower monomers. Helix axes in the right-most monomer are dashed. (E) Details of residues that support the structure of loops 1 and 2. (F) Simulated annealing composite omit density (mesh) contoured at 1σ around W193. (G) Detail of D2.2 showing Arg185 in its carbonyl cage, including the sidechains of E126 and Q379 and backbone carbonyl groups of residues 382 and 389. (H) Orientations of D2.2 of the 3.3 Å (light blue) and 2.6 Å (green) HAP2 structures after superposition on D1, D2.1, and D3 in the trimer.

DOI: https://doi.org/10.7554/eLife.39772.012

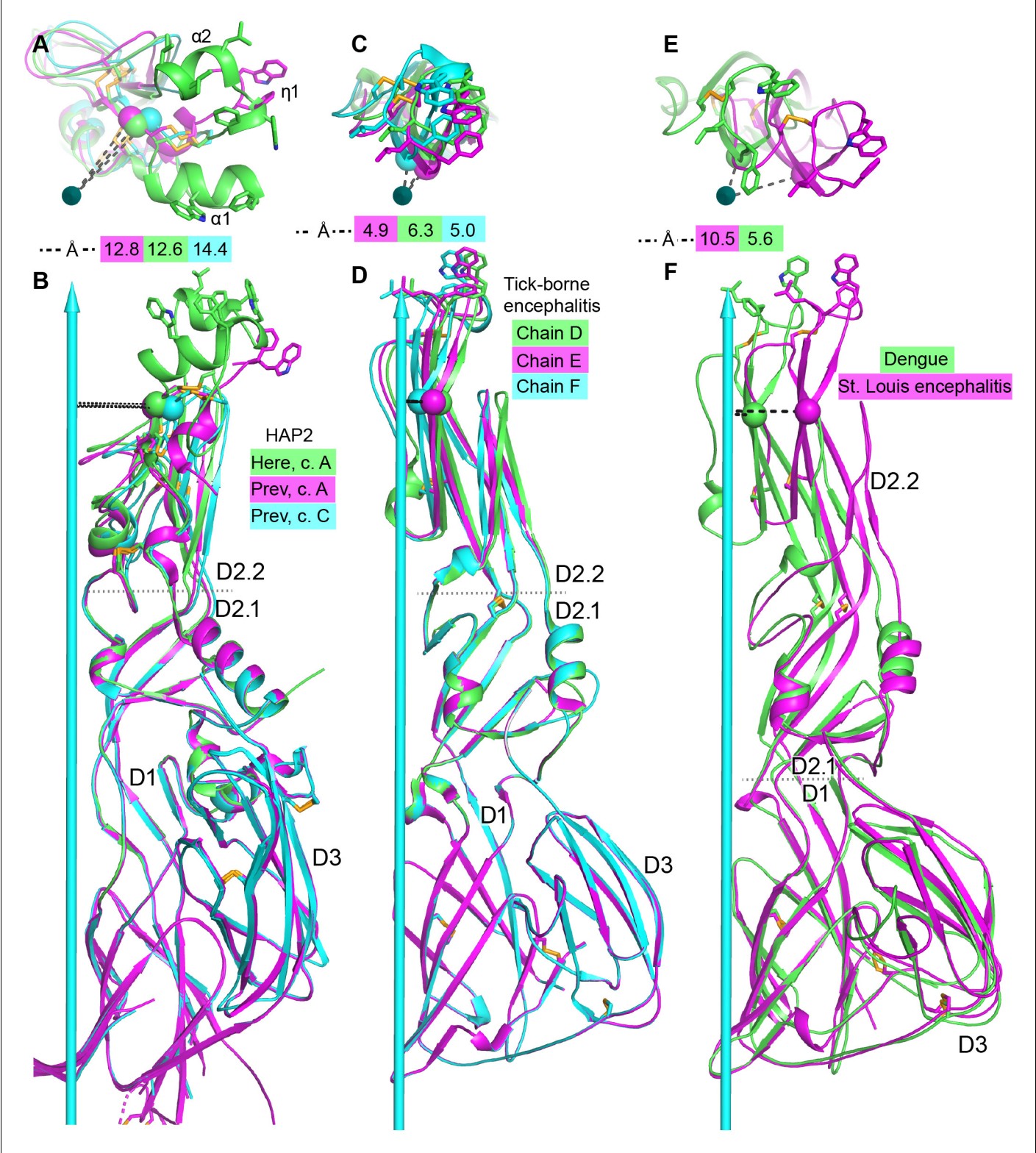

**Figure 5.** Comparisons of tilting. (**A, B**) HAP2 from the current structure (green) and chains A (magenta) and C (cyan) from the previous structure (*Fédry et al., 2017*). (**C, D**) *Tick-borne encephalitis* fusogen chains D, E, and F. (*Bressanelli et al., 2004*). (**E, F**). *Dengue* 1 (*Nayak et al., 2009*) (green) and *St. Louis encephalitis* (*Luca et al., 2013*) (magenta) fusogens. Distances from a conserved D2.2 framework Cα atom (residue 166 in HAP2 and 96 in the flaviviruses) to the 3-fold axis are dashed and shown in the key. Dotted lines mark boundaries at which tilting occurs. Views are straight down the 3-

*Figure 5 continued on next page*

*Figure 5 continued*

fold axis shown as a cyan arrow (**A, C, E**) or normal to the axis (**B, D, F**). Structures are shown in ribbon cartoon with disulfide bonds and key fusion loop sidechains in stick. Monomers are superimposed on D1.

DOI: https://doi.org/10.7554/eLife.39772.013

comparable in extent (up to ~2 Å) and in exhibiting both radial and circumferential motion. Thus, the HAP2 monomer in cyan and the *Tick-borne encephalitis* monomer in green differ from their counterpart monomers radially. In contrast, the HAP2 monomers in magenta and green and the *Tick-borne encephaliti*s monomers in cyan and magenta differ from one another in circumferential position (*Figure 5A–D*). Among flavivirus fusion state structures, perhaps the largest difference in orientation at the D1-D2.1 junction is seen between *Dengue* 1 and *St. Louis encephalitis* flavivirus fusogens (*Figure 5E,F* and *Figure 6*). Viral class II fusogens also flex at the D1-D2.1 junction between their pre-fusion and fusion states (*Harrison, 2015*; *Kielian and Rey, 2006*) and to a lesser extent at the D2.1-D2.2 junction. Flexibility visualized in comparison between flavivirus fusion state structures has both radial and circumferential components (*Figure 5E–F*) and is larger in extent than seen in the examples of D2.1-D2.2 flexion with HAP2 and *Tick-borne encephalitis* (*Figure 5A–D*). Flexibility at both junctions, D2.1—D2.2 and D1–D2.1, has implications for flexibility of type II fusogens in general and for HAP2 (Discussion).

Additional marked differences in the fusion loops between the current and previous HAP2 structures are independent of D2.1–D2.2 flexion. Of the fusion loop segment from residue 167 to 204, only fragments from residue 182 to 194 (monomer A) or from residue 184 to 190 or 191 (monomers B and C) were built in the previous structure. Moreover, the entirety of these segments differs from that in our structure. The segment from Arg185 to Cys190 in the 2.6 Å structure is contracted in the 3.3 Å structure to form an α-helix (α+ in *Figure 4H*). The difference in conformation and the contraction might result from protease cleavage of the loops flanking this segment in the previous structure. Whatever the cause, fusion loop residues Phe192 and Trp193, present in only monomer A in the 3.3 Å structure, differ in position by 6 and 9 Å, respectively, from our structure. Although the

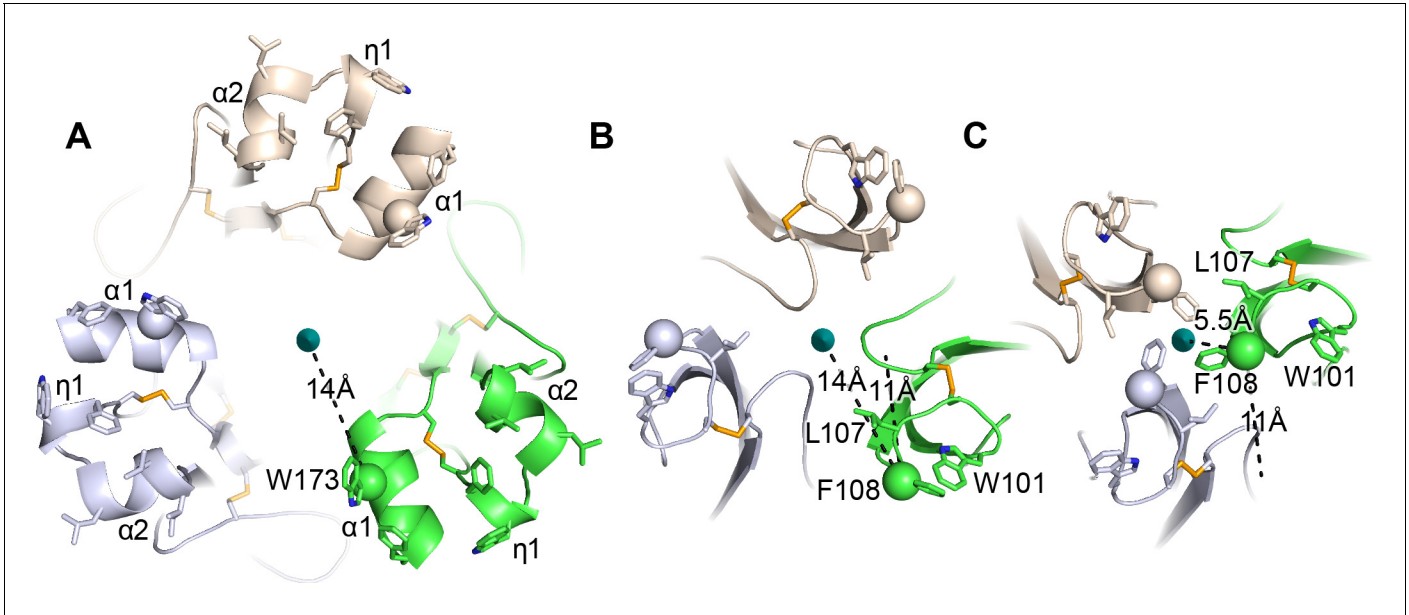

**Figure 6.** HAP2 fusion loops and comparison to iris-like movements of flavivirus fusion loops. (**A**) Current HAP2 structure. B and C. Flavivirus fusogens from *St. Louis encephalitis* (**B**) and *Dengue 1* (**C**) viruses. Trimers were superimposed on a common 3-fold axis using D1 and equivalent monomers are colored identically. Dashes show distances in one monomer from Cβ atoms of the indicated fusion loop residues to the 3-fold axis and also, in (**B and C**), distances between Cβ atoms in the two flaviviral fusogen structures. From (**B to C**), tilting at the D1—D2.1 junction shown in *Figure 5* results in iris-like movement of the fusion loops in a counterclockwise and axis-proximal direction.

DOI: https://doi.org/10.7554/eLife.39772.015

conformation of the fusion loop fragments in the previous structure differs from that in ours, these fragments are held in roughly the same position by binding of Arg185 to its carbonyl cage and disulfide linkage of Cys190 to Cys167 (Arg185 and Cys 190 are included in the shortest of the fragments previously traced, from residues 184 to 190). Interestingly, between the two structures, the Cα atom of Cys190 differs more in position (4.2 Å) than the central carbon of Arg185's guanido group (1.1 Å) or its Cα atom (2.4 Å), showing that this segment of the loop is anchored in position more by Arg185 than by Cys190. Thus, positioning of the Arg185 sidechain in the carbonyl cage of the D2.2 subdomain is robust to substantial movements in the loop it subtends.

HDX provided further evidence for differential mobility within the two fusion loops. Peptide 173–188 containing Arg185 and a portion of fusion loop 1 with the α1-helix was in rapid exchange (*Figure 2A–C*). Moreover, the backbone amide hydrogens in this fusion loop 1 peptide exchanged more rapidly in trimeric than monomeric HAP2 (*Figure 2D*). These results are consistent with the paucity of backbone hydrogen bonds in the non-helical portion of this segment and its exposure to solvent in the trimeric structure. In contrast to fusion loop 1, fusion -loop 2 peptides 192–198 and 193–200 containing the η1 and α2-helices exchanged with only moderate kinetics (*Figure 2A–B*), and exchange was decreased in the trimer compared to the monomer (*Figure 2D*). The kinetics of exchange in each overlapping peptide showed two distinct groups of residues, with one group of residues exchanging from 0 to 10 min, and another group not exchanging from 10 to 300 min and thus highly stable (*Figure 2—figure supplement 2D-E*). Therefore, at least a portion of the loop two sequence $W^{193}SDPLDIL^{200}$, which contains three of five loop two residues implicated in fusion, has a stable structure in both the monomer and trimer. The increase in loop one and decrease in loop two dynamics suggest that both regions alter in structure or exposure upon trimer formation.

Surprisingly, secondary structures in HAP2 domains D1, D2.2, and D3 that are closer to the 3-fold axis, and more buried between protomers of the trimeric structure, tend to exchange more rapidly than regions that are exposed on the trimer perimeter (*Figure 2A,B*). In D1, the A0B0I0H0G0 β-sheet faces the trimer axis (*Figure 5A*), and exchanges more rapidly than the J0C0D0E0F0 β-sheet that is exposed to solvent on the perimeter of the trimer (*Figure 5B*). In D3, the edge of the β-sandwich faces the trimer axis. Axis-proximal β-strands B, C, D, and E exchange more rapidly than axis-distal β-strands A and G. In D2.2, the exposed b and d β-strands were in slow exchange. Furthermore, the c β-strand and adjacent loops, which are close to the 3-fold axis, exchanged more rapidly. Similarly, the axis-proximal α3-helix was in moderately fast exchange. These HDX results show that in most domains, regions in trimer interfaces are in more rapid exchange. Rapid exchange, which largely correlates with flexibility, may be a specialization that permits reshaping during monomer to trimer transition, and alterations in D2.1 and D2.2 orientation in fusion-state structures.

## Hydrophobic residues in the axis-proximal α1- and α2-helices are more important in fusion than in the axis-distal η1 helix

Our finding that the long fusion loop of HAP2 displayed three distinct helices, each projecting sets of hydrophobic residues that could interact with the lipid bilayer of the target membrane, was unexpected. This finding provided the opportunity to test the hypothesis that the residues in each helix had an equivalent functional effect on *Chlamydomonas* gamete fusion. The Trp and Phe residues in the α1 and η1-helices are the most hydrophobic aromatic residues and the Leu and Ile residues in the α2-helix are the most hydrophobic aliphatic residues. We examined the functional relevance of these hydrophobic residues by mutating them to Ala. We thus tested whether fusion of a *Chlamydomonas hap2* mutant could be rescued by transformation with hemagglutinin (HA)-tagged HAP2 (HAP2-HA) transgenes bearing mutations in the α1-helix (W173A/F177A, *fh1*), η1-helix (F192A/W193A, *fh2*), or α2-helix (L197A/L200A/I201A, *fh3*).

HAP2 protein is expressed only when vegetatively growing *minus* cells are induced to become gametes. We first established proper surface localization in *hap2 minus* gametes. Immunoblotting showed the typical HAP2 doublet with a larger surface-expressed form and a smaller, intracellular form (*Liu et al., 2015*) in *hap2* gametes expressing wild-type and all three mutants forms of HAP2-HA (*wt-, fh1-, fh2-*, and *fh3*), and as expected, not in wild-type *plus* gametes (*wt+*) (*Figure 7A*). Furthermore, trypsin-treated gametes lost the upper but not lower HAP2 band (*Figure 7B*). HAP2 is localized in *minus* gametes to a small patch of membrane, the *minus* mating structure, between the two cilia. Anti-HA immunofluorescence combined with differential interference contrast (DIC) microscopy showed that wild-type and each mutant HAP2 were present between the two cilia at the site of

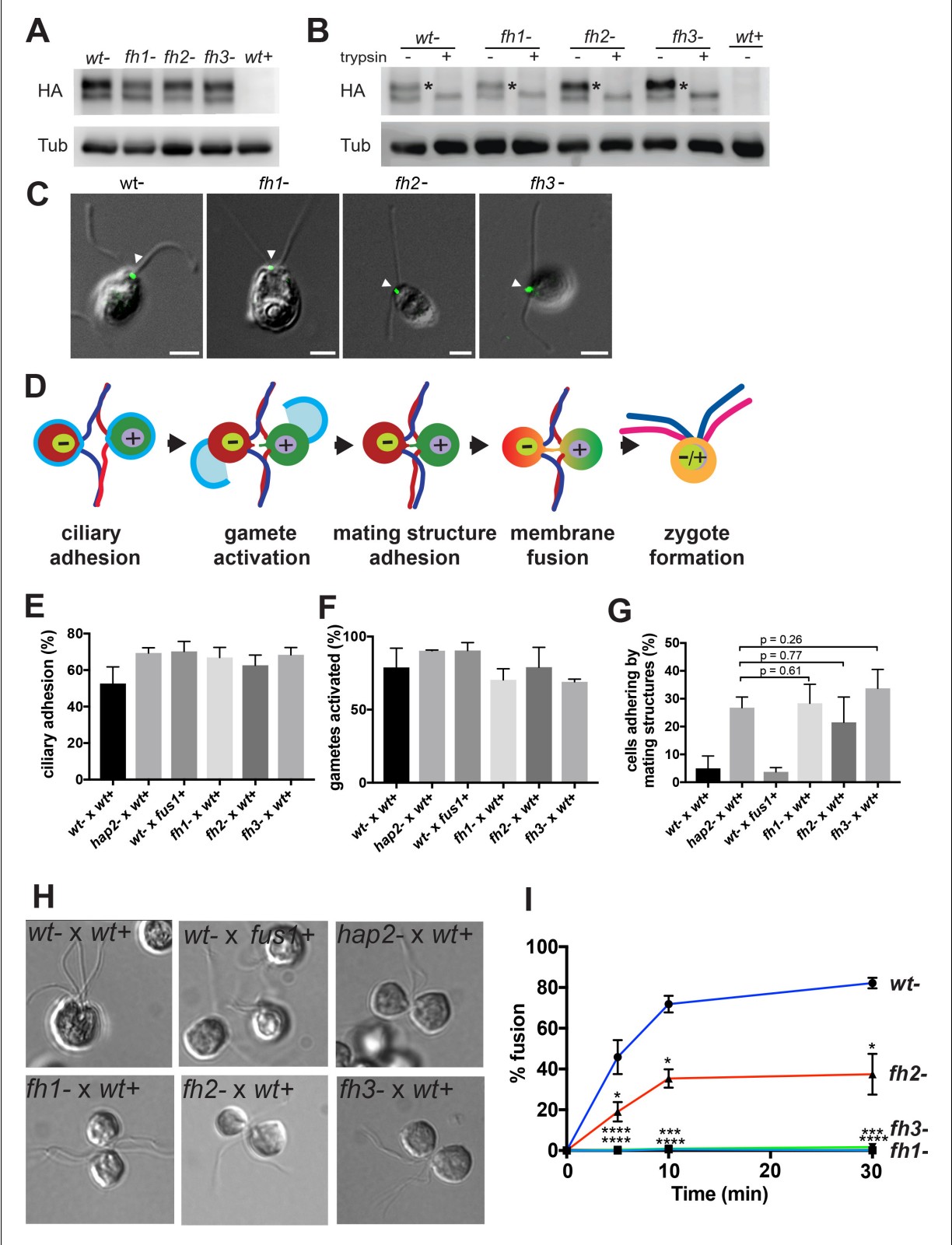

**Figure 7.** Mutations in three distinct HAP2 fusion loop helices specifically disrupt the membrane fusion step in *Chlamydomonas* fertilization. (**A**). Protein expression and (**B**) trypsin-sensitivity of HA-tagged wild type HAP2 and fusion helix mutants in *minus* gametes detected by immunoblotting with anti-HA (upper). *Plus* gametes (*wt+*) are used as a negative control. Blotting with anti-tubulin (lower) controlled for loading. (**C**) Combined immunofluorescence staining with anti-HA and differential interference contrast (DIC) microscopy show the location of wild type and HAP2 fusion helix

*Figure 7 continued*

mutant proteins on the *minus* mating structure between the two cilia (arrow heads). (D) Schematic illustration of steps (left to right) in *Chlamydomonas* fertilization. (E-I) Assays with the indicated mixtures of *minus* and *plus* gametes. (E) Ciliary adhesion at 5 min after mixing as assessed by particle counting. Because over 40% of the cells in the *wt*- x *wt*+ sample fused by 5 min, and because ciliary adhesion is downregulated upon fusion, the *wt*- x *wt*+ sample exhibited slightly less ciliary adhesion than the mutants. Analysis by the Kruskal-Wallis and Dunn's post tests showed no significant differences among the mutants. (F) Gamete activation as assessed by cell wall loss. A Kruskal-Wallis test showed no significant difference in gamete activation among samples. (G and H). Mating structure adhesion. (G) Quantification. Dunn's post test showed no significant difference in mating structure adhesion between *hap2* and *fh1*, *fh2*, and *fh3* gametes mixed with *wt*+ samples. Negative controls were *wt*- x *wt*+ gametes, which fuse too quickly to detect mating structure adhesion; and *fus1* mutants, which lack the *plus* gamete adhesion protein and so are incapable of mating structure adhesion. (H) Representative DIC images showing that *wt*- x *wt*+ gametes form zygotes; *wt*- x *fus1*+ fail to form pairs; and *hap2*-, *fh1*-, *fh2*, and *fh3*- all form pairs with *wt*+. (I). Effect of mutating the fusion helices' hydrophobic residues on fusion as measured by the formation of quadri-ciliated zygotes. In (E–G) and (I), values shown are averages from at least two biological samples with three different replicates each; results are shown as mean ± SD of all replicates. In (I), levels of pairwise significance (Dunn's post test) between the *wt*- control and *fh1*-, *fh2*-, and *fh3*- samples at 5, 10 and 30 min after mixing gametes are $p<0.05$, *; $p<0.001$,***; $p<0.0001$, ****.

DOI: https://doi.org/10.7554/eLife.39772.014

the *minus* mating structure (*Figure 7C*). Thus, the three fusion helix mutations gave rise to HAP2 proteins that were of the correct size in SDS-PAGE, surface-expressed as shown by trypsin susceptibility, and localized to the mating structure as shown by microscopy.

Gamete fusion in *Chlamydomonas* is the culmination of a series of complex cellular events initiated when *plus* and *minus* gametes are mixed together (*Figure 7D*). Successful gamete fusion, which occurs within minutes after gamete mixing, first requires that cells undergo ciliary adhesion that results in cellular agglutination. Ciliary adhesion-induced gamete activation then elicits release of cell walls and formation of membrane protuberances, the activated *plus* and *minus* mating structures. Finally, the tips of these mating structures adhere to each other, followed by bilayer merger and complete cell coalescence to form a quadri-ciliated zygote.

We established that expression of the *fh* HAP2 transgenes was without effect on these pre-fusion steps. Cellular agglutination as measured by electronic particle counting showed that *fh minus* gametes were indeed as capable of ciliary adhesion as wild-type *minus* gametes when mixed with *plus* gametes (*Figure 7E*). In the next step of fertilization, *plus* and *minus* gametes lose their cell walls (*Figure 7D*). Cell wall loss, as measured by the susceptibility of wall-less but not walled gametes to mild detergent-mediated release into the supernatant of cytoplasmic contents including OD435-detectable chlorophyll, was robust and indistinguishable from wild-type in *fh* mutants (*Figure 7F*), showing that they became activated. The prelude to fusion is mating structure adhesion (*Figure 7D*), which was assayed by subjecting mixtures of *plus* and *minus* gametes that had passed the activation step to fixation and strong pipetting to disrupt ciliary adhesion, and then counting the % of cells that remained as pairs adhering only by their mating structures. All three *fh* mutants were competent in mating structure adhesion (*Figure 7G and H*). Only a few pairs were found in the wt- x wt+ cross (*Figure 7G*), owing to the rapid fusion and formation of quadri-ciliated zygotes exhibited by the *hap2 minus* gametes expressing wild-type HAP2-HA (*Figure 7H* first panel). Notably, mating structure adhesion, but not any earlier step, was abolished by the use of *fus1 plus* gametes, which lack an adhesion molecule required on *plus* gametes for mating structure adhesion (*Misamore et al., 2003*) (*Figure 7E–G*).

Finally, having established that HAP2 *fh* mutants were competent in all steps in fertilization leading up to fusion, we examined fusion itself. *Plus* gametes were mixed with equal numbers of *hap2 minus* gametes expressing wild-type or *fh* mutant forms of HAP2-HA and fusion was assayed as the percent of gametes that had progressed from having two cilia to zygotes with four cilia (*Figure 7I*). Whereas fusion with wild-type HAP2-HA gametes was nearly 50% after 5 min and over 80% by 30 min, fusion was essentially absent in the *fh1* and *fh3* mutants (maximal fusion was <1% in fh1 and <2% in fh3). Fusion with *fh2* gametes was significantly reduced and less than half of wild-type at all time points. These results demonstrate that the apically exposed, hydrophobic residues in each of the helices in the HAP2 fusion loops are indeed important in cell fusion during *Chlamydomonas* fertilization. Furthermore, W173 and F177 in the α1-helix mutated in *fh1* and L197, L200, and I201 in the α2-helix mutated in *fh3* were essential for fusion, while F192 and W193 in the more distal η1-helix mutated in *fh2* were only moderately, albeit significantly, required for fusion.

Thus, the mutational results showed that residues in the α1 and α2-helices, which are proximal to the trimer 3-fold axis, have an essential role in fusion, while residues in the η1-helix, which are distal from the trimer 3-fold axis, are only marginally important in fusion. Below, we interpret our structural and functional results, and structures of viral fusogens such as those shown in *Figures 5* and *6*, in support of a model in which the fusion loops in each HAP2 monomer approach the 3-fold axis in the fusion state interrogated in vivo by mutation. In this model, the HAP2 crystal structure represents an intermediate state in fusion. Pivoting or iris-like movements in D2.1 and/or D2.2 during the fusion reaction result in a unitary fusion surface with essential, axis-proximal α1- and α2-helices at the center of a membrane interaction surface, and less-necessary, axis-distal η1 helices at the periphery.

## Discussion

We have characterized the crystal structure of HAP2 from *Chlamydomonas reinhardtii* in a trimeric fusion state at 2.6 Å, the dynamics of its polypeptide backbone in both monomeric and trimeric states, and the functional importance in gamete fusion of residues in its fusion loop. We were fortunate that our structure allowed us to completely trace the unusually long fusion loop in *Chlamydomonas* HAP2, revealing that the hydrophobic residues in its 2 loops are apically exposed on 3 helices.

Our structure also revealed other long loops in HAP2, which altogether make its sequence from D1 to D3 (560 residues in total) longer than in previously crystallized class II fusogens from viruses (391 to 431 residues) or a *C. elegans* somatic cell fusogen (509 residues) (*Pérez-Vargas et al., 2014*). HAP2 also has substantially more N-glycosylation sites, and longer putatively O-glycosylated Ser, Thr, and Pro-rich sequences, than viral class II glycoproteins. It is possible that these long loops and glycans stabilize HAP2, or the fusing membrane bilayers, by decreasing the volume accessible to alternative disordered protein or membrane states. They may therefore have functional significance. An alternative explanation is that the requirement for class II viral fusogens to pack tightly against one another and other proteins in an icosahedral lattice on viral surfaces may select against long loops and bulky glycans, whereas HAP2 should lack similar evolutionary constraints.

### HAP2 dynamics and flexibility within D2

HDX provided insights into the backbone dynamics of HAP2 and differences between its monomeric and trimeric states. Slower exchange at multiple domain-domain interfaces in the trimeric state compared to the monomeric state was consistent with less exposure or structural rearrangements upon trimer formation. Regions of slower exchange in trimers included interfaces buried on D3, and buried on D1 and D2.1, where D3 packs against D1 and D2.1 in trimers, and thus provided evidence for a conformational transition from a more linear arrangement of these domains in monomers that was altered to the arrangement with the hairpin bend between D1 and D3 in trimers. Conversely, slower exchange in the monomeric state in a loop in D3 that is exposed in the trimeric state suggested that this loop could contact D1 in a more linear arrangement of domains in the monomer. Large HDX differences were also found in fusion loops 1 and 2 between the trimeric and monomeric forms, showing that these loops are capable of structural alterations or differ in exposure in these states. Importantly, in both the monomeric and trimeric states of HAP2, its fusion loops were highly to moderately dynamic. The loop that separates the two fusion loops, with Arg185 at its tip, was especially dynamic.

Comparisons between HAP2 structures show that D2.2 can tilt at its junction with D2.1, and that the fusion loops can rock at Arg185. The D2.1—D2.2 junction is thin, with only four polypeptide connections, none of which contain canonical secondary structure. Comparison between the two structures shows considerable displacement of the loop with Arg185 at its tip and conformational change within the portion of this loop resolved in the previous HAP2 structure, despite little displacement of the Arg185 guanido group relative to its carbonyl cage. Carbonyl cages support rocking motion of the PSI domain relative to the hybrid domain in integrins and allosteric motion in selectins at the lectin-EGF domain interface (*Springer, 2009*; *Xia and Springer, 2014*). In HAP2, HDX showed that the loop with Arg185 at its tip was in rapid exchange, while the carbonyl cage exchanged slowly. Slow exchange of the cage is consistent with its construction from secondary structure elements that have many other contacts and hydrogen bonds in the structure, while the long loop with Arg185 at its tip has few significant contacts or hydrogen bonds except to itself. Interestingly, PFAM 10699 forms

much of the carbonyl cage and provides three of the four residues that hydrogen bond to Arg185 (*Figure 4C*). Therefore, one of the drivers for conservation of this segment appears to be the carbonyl cage.

## A model for merger of the fusion loops in each HAP2 monomer to form a large, unified fusion surface

We now discuss evidence that suggests that our HAP2 crystal structure is an intermediate state, and that in the state interrogated by mutagenesis, the fusion loops merge at the 3-fold trimer axis to form a large, unified fusion surface. According to current models for viral class II fusion protein conformational change, trimers pass through early and late fusion state structures on the pathway to membrane fusion (*Harrison, 2015*; *Klein et al., 2013*). Pivoting at the D1—D2.1 interface is well described in the literature, both between fusion state structures, and between pre-fusion and fusion state structures. The case for a continuum in D2 or D2.2 orientation is especially well made by the *Dengue 1*, *Tick-borne encephalitis*, and *St. Louis encephalitis* flaviviral fusogens. Their fusion-state structures show essentially identical conformations in D1 and D3; in contrast, D2 orientation varies markedly (*Figures 5* and *6*). Thus, *Dengue 1* fusion loops are in symmetric, intimate van der Waals contact at the 3-fold axis (*Nayak et al., 2009*), *Tick-borne encephaliti*s fusion loops are in asymmetric, less intimate contact (*Bressanelli et al., 2004*), and *St. Louis encephalitis* fusion loops are distal from the 3-fold axis (*Luca et al., 2013*). Notably, these fusogens are 36, 42, and 49% identical in sequence to one another, and have identical fusion loop residues. Because these flaviviral fusogens are closely related in sequence and function, it is reasonable to hypothesize that each passes through a continuum of conformations similar to that exhibited by the entire set of structures. We think of these flavivirus fusogens as homologues of the same protein in different species, just as we think of HAP2 as the same protein in different species; indeed, these flaviviral fusogens are much more closely related to one another than HAP2 fusogens from different species, which show only 12% to 35% sequence identity for the representatives in (*Figure 3—figure supplement 1* and *2*) and vary greatly in their fusion loops. Thus, flaviviral fusion state crystal structures appear to represent different frames in a movie that plays similarly for all flavivirus fusogens, and by extension, for other structurally and functionally homologous class II fusion proteins such as HAP2.

Our in vivo analysis of *Chlamydomonas* HAP2 mutants showed that the hydrophobic residues in the fusion loops are important during the fusion reaction, consistent with a role in interaction with the membrane of the *plus* gamete. Moreover, the three topographically distinct sets of fusogenic residues in each HAP2 monomer provided a unique opportunity to test their relative importance both within the pre-fusion state of HAP2 on resting gametes and during the fusion process per se. None of the mutations in the fusion loop helices had any detectable effect on expression or localization of HAP2 in resting gametes, indicating that these residues have little if any function before the fusion reaction begins. On the other hand, mutation of residues in the α1 and α2-helices decreased fusion efficiency by 50 to 100-fold, while mutations in the η1-helix decreased fusion efficiency by only 2-fold. Previous mutation of the η1-helix yielded identical results (*Fédry et al., 2017*). Notably, mutation of identical residues, Phe and Trp, in the α1 and η1-helices yielded markedly different results, showing that differences in location, rather than character of the mutated residues, was responsible for the differences in phenotype.

Several models potentially could explain our overall findings. One model proposes that the current crystal structure mediates fusion, and that each monomeric D2.2 associates independently with the target membrane. Another model, proposed with a viral fusogen splayed more than here, is that splayed trimers could associate laterally to form rosettes, in which the outer faces of D2.2 associate to form trimer-trimer associations (*Gibbons et al., 2004*). A third model proposes that in the final fusion state, the inner, 3-fold axis proximal face of D2.2 converges on the 3-fold axis as seen in *Dengue* (*Harrison, 2015*; *Nayak et al., 2009*; *Klein et al., 2013*). The fusion process is highly cooperative, and thus all models postulate that multiple trimers nearby one another are required for fusion.

We test these models with a concept from structural biology in which it has been found that the most important residues for protein interactions are predominantly near the center of interaction surfaces, that is in hotspots (*Clackson and Wells, 1995*; *Bogan and Thorn, 1998*). In the model in which monomers are sufficient, the fusion loops in each monomer are separated from one another by solvent, and function independently of one another. In monomers, the least important η1 helix lies in between the most important α1 and α2-helices; therefore, the hotspot model is not consistent

with the monomers functioning separately in the state most mutationally important for fusion. In the model in which trimers associate laterally into rosettes, association occurs through the monomer face distal from the 3-fold axis that bears the η1 helix (*Gibbons et al., 2004*). This helix is the least important in fusion, and thus association through rosettes in the state interrogated by mutagenesis is not consistent with the hotspot concept. In the model in which the monomers close up at the 3-fold axis, the α1 and α2-helices on neighboring monomers would approach one another and be central in a fusion surface while the η1-helix would be on the periphery. A single, trimeric fusion surface is consistent with the hotspot model, since the α1 and α2-helices at the center of the membrane interaction surface are critically important for fusion, while the η1-helix on the outer periphery of the interface is only marginally important. Thus, in the most mutationally important state for fusion, the hotspot concept argues against the monomers being splayed apart and acting independently or associating into rosettes, and argues for merger of the monomers at the three-fold axis into a single fusion surface. The mutational results thus suggest that our crystal structure does not represent the final fusion state of HAP2 during the fusion reaction, but instead is one frame in a movie of a model of HAP2 conformational change during the gamete fusion process.

Our results do not argue against a role during a multi-step fusion process for interaction of splayed trimers or rosettes of splayed trimers with the target membrane. Instead, the results argue for approach of the α1 and α2-helices to the 3-fold axis in the state that is most critical for membrane fusion, which is likely to correspond to the transition state for membrane fusion. Transition states are by definition the highest energy states during reaction processes, and thus the states most in need of structural stabilization. Correspondingly, we propose that the transition state is the most sensitive state to mutation, and therefore that the fusion surfaces of each monomer move towards the 3-fold axis to form a single, large fusion surface in the final state for membrane fusion, which would also likely be the state in which the fusion loops are most deeply buried in the membrane.

Our structural and dynamics studies on HAP2 and analysis of flaviviral fusion state structures provide plausible pathways by which D2.2 in each monomer can approach the 3-fold axis to form a common fusion surface. From *Figure 6B* to *Figure 6C*, the flaviviral fusion loops in each monomer move counterclockwise by 11 Å at Phe108 and, like a closing iris, come 8.5 Å closer to the 3-fold axis and into van der Waals contact. This movement is accomplished by tilting at the D1—D2.1 junction (*Figure 5E,F*). The D2.2 framework in our HAP2 structure is at a similar distance from the 3-fold axis and has a similar D1—D2.1 tilt as the more distal flaviviral structure (*Figure 5B,F*), resulting in the similar positions at 5 o'clock of the green monomer fusion loops in *Figure 6A,B*. Similar movement at the D1–D2.1 junction in HAP2 is plausible, which would bring the green monomer fusion loops in *Figure 6A* closer to the 3-fold axis and toward the 3 o'clock position as seen in *Figure 6C*. Note that in iris-like motion, W101 in flaviviral fusogens remains distal from the 3-fold axis (*Figure 6B,C*), as would the η1-helix in HAP2 (*Figure 6A*). HAP2 and flaviviral fusogens also show tilting at the D2.1—D2.2 junction (*Figure 5A–D*). Moreover, crystal structure comparisons show and HDX suggests that the HAP2 fusion loops can rock at the carbonyl cage. We expect that tilting at the D1—D2.1 junction resulting in iris-like motion of fusion loops, tilting at the D2.1—D2.2 junction, and rocking of fusion loops in the carbonyl cage may all contribute to the approach toward the 3-fold axis of the fusion loops in HAP2. The orientation with respect to the 3-fold axis of the three, hydrophobic residue-bearing helices within each monomer of HAP2 are likely to change little during movement toward the 3-fold axis, as seen in flavivirus fusogens. The α1-helix of one monomer would come into contact with the α2-helix of its neighbor, whereas the η1-helix would form the outer edge on the perimeter of the merged trimeric fusion surface.

In summary, our structure of a trimer of the *Chlamydomonas* HAP2 ectodomain shows that the hydrophobic residues in the fusion loops of this class II fusogen project apically from three helices at the tip of D2.2. From the perspective of the hotspot model for protein interactions, the splaying of the fusion loops from the 3-fold axis of the trimer along with the lesser importance of the most distal, η1-helix make it likely that our structure does not represent the final fusion state of HAP2. Rather, flexibility within D2 uncovered by protein dynamics using HDX along with comparisons among HAP2 structures and to flaviviral fusogens lead to the model that HAP2 fusion loops in each monomer merge together to form a larger, unified fusion surface during the final stages of the fusion mechanism.

# Materials and methods

## Key resources table

| Reagent type (species) or resource | Designation | Source or reference | Identifiers | Additional information |
|---|---|---|---|---|
| Gene (*Chlamydomonas reinhardtii*) | *HAP2* | NA | JGI:Cre16. g674852.t1.1; NCBI:EF397563 ;Uniprot ID: A4GRC6 | |
| Gene (*Chlamydomonas reinhardtii*) | *FUS1* | NA | JGI:Cre06. g252750.t1.1; NCBI:U49864 | |
| Genetic reagent (*Chlamydomonas reinhardtii*, mt+) | *wt+ (also called 21gr)* | Chlamydomonas Resource Center PMID: 17247567 | CRC:CC-1690 | Dr. Ruth Sager (The Sidney Farber Cancer Institute, Boston, MA, December 1983). |
| Genetic reagent (*C. reinhardtii*, mt+) | *fus1+; (also called fus1-1)* | Chlamydomonas Resource Center PMID: 8856667 | CRC:CC-2062 | Dr. Ursula Goodenough (Washington University, St. Louis, MO, December 1986). |
| Genetic reagent (*C. reinhardtii*, mt-) | *hap2- (also called 40D4)* | Chlamydomonas Resource Center PMID: 25655701 | CRC:CC-5281 | Dr. Yanjie Liu (UT Southwestern Medical Center, Dallas, TX, May 2016); Dr. William Snell; NIT plasmid (pMN56) transformed into progenitor strain B215 |
| Genetic reagent (*C. reinhardtii*, mt-) | *wt-* | this paper | | The HAP2-HA plasmid (pYJ76) was modified by removal of two internal residues used for the plasmid's generation to form a new plasmid pyJJ1. This was transformed into *hap2-* |
| Genetic reagent (*C. reinhardtii*, mt-) | *fh1-* | this paper | | A W173A and F177A modified HAP2 plasmid (pYJJ2) was transformed into *hap2-* |
| Genetic reagent (*C. reinhardtii*, mt-) | *fh2-* | this paper | | F192A and W193A modified HAP2 plasmid (pYJJ3) was transformed into *hap2-* |
| Genetic reagent (*C. reinhardtii*, mt-) | *fh3-* | this paper | | L197A, L200A, and I201A modified HAP2-HA plasmid (pYJJ4) was transformed into *hap2-* |
| Strain, strain background (*Escherichia coli*) | DH5α | thermo fisher scientific | Catalog NO: 18265017 | |
| Genetic reagent (*Escherichia coli*) | QIAquick Gel Extraction Kit | Qiagen | Catalog NO: 28704 | https://www.qiagen.com/ |
| Genetic reagent (*Escherichia coli*) | QIAGEN Plasmid Mini Kit | Qiagen | Catalog NO: 12123 | https://www.qiagen.com/ |
| Genetic reagent (*Drosophila*) | EXpreS2 transfection reagent | ExpreS2ion Biotechnologies | Catalog NO: 95-055-075 | https://expressionsystems.com/product/expres2-tr-transfection-reagent/ |

*Continued on next page*

*Continued*

| Reagent type (species) or resource | Designation | Source or reference | Identifiers | Additional information |
|---|---|---|---|---|
| Recombinant DNA reagent (plasmid) | pYJ76 | PMID: 25655701 | | original HAP2-HA plasmid for transformation of *Chlamydomonas* |
| Cell line (*Drosophila*) | *Drosophila melanogaster Schneider* S2 | ExpreS2 cells | ExpreS2ion Biotechnologies | |
| Transfected construct (*Drosophila*) | ET15S2 vector | This paper | ExpreS2ion Biotechnologies | Modified with the pExpreS2-2 vector;Includes N-terminal secretion signal from *Hspa5* and C-terminal His8 tag |
| Biological sample () | N/A | | | |
| Antibody | HA antibody (Rat monoclonal) | Sigma | clone 3F10 | WB(1/1000); IF(1/100) |
| Antibody | Alexa Fluor 488- goat anti-rat secondary | Invitrogen | | (1/400) |
| Antibody | Goat Anti-Rat IgG Peroxidase | Millipore | | (1/5000) |
| Commercial assay or kit | Poroszyme Immobilized Pepsin Cartridge, 2.1 mm x 30 mm | Applied Biosystems | Catalog NO: 2313100 | HAP2 digestion for HDX experiment |
| Chemical compound, drug | Polyethylene glycol 3350 | Hampton research | Catalog NO: HR2-527 | For HAP2 crystallization |
| Chemical compound, drug | n-Dodecyl-β-D-maltoside (DDM) | Sigma-Aldrich | Catalog NO: D4641-5G | For HAP2 trimerization |
| Chemical compound, drug | Ammonium acetate | Hampton research | Catalog NO: HR2-565 | For HAP2 crystallization |
| Software, algorithm | Prism 7 | https://www.graphpad.com/scientific-software/prism/ | | GraphPad software used for statistical analyses |
| Software, algorithm | XDS | https://strucbio.biologie.uni-konstanz.de/xdswiki/index.php/Xds | | Diffraction data was processed with XDS |
| Software, algorithm | Phenix | https://www.phenix-online.org/ | | The structure was solved by molecular replacement with Phaser in the Phenix suite |
| Software, algorithm | CCP4 | http://www.ccp4.ac.uk/ | | Refinement |
| Software, algorithm | ASTRA 6 | https://www.wyatt.com/products/software/astra.html | | SEC-MALS data were processed in ASTRA six using the protein conjugate model |
| Software, algorithm | DynamX 3.0 | Waters Corp. | http://www.waters.com/waters | HDX |
| Software, algorithm | PLGS 3.0 | Waters Corp. | http://www.waters.com/waters | HDX |

## Expression and purification of HAP2 ectodomain

cDNA encoding *Chlamydomonas* HAP2, residues 23–582, codon optimized for mammalian cell expression, was cloned into ET15S2 vector, a ligation-independent cloning variant of the pExpreS2-2 vector (ExpreS2ion Biotechnologies) that includes N-terminal secretion signal from *Hspa5* and

C-terminal His8 tag. *Drosophila melanogaster Schneider* S2 cells (ExpreS2 cells, ExpreS2ion Biotechnologies), grown in EX-CELL 420 Serum-Free Medium (Sigma), were transfected using EXpreS2 transfection reagent. Stable transfectants were selected in the same medium supplemented with 4 mg/ml G418 and expanded in EX-CELL 420 medium at 25°C. After centrifugation at 5,000 g for 20 min, 1 L culture supernatant was filtered (0.22 µm pore) and made 2 mM in $NiCl_2$ and 300 mM in NaCl. Protein was purified using a 10 ml $Ni^{2+}$-nitrilotriacetate column (Qiagen) followed by size exclusion chromatography using a Superdex 200 10/300 GL column in 20 mM Tris-HCl, pH 7.5, 500 mM NaCl with a yield of 1 – 1.5 mg per L supernatant.

## Crystallization and structure determination

Crystals were grown at 20°C by hanging-drop vapor diffusion with equal volumes of protein (2.5 mg/ml) and reservoir solution, 26% polyethylene glycol 3350, 0.1 M HEPES pH 7.5, 0.35 M ammonium acetate. Hexagonal plate crystals were cryo-protected with reservoir solution containing 15% glycerol in 2 steps of 5% and 10% increase and plunged in liquid nitrogen. Data were collected at 100° K on GM/CA beamline 23IDB at the Advanced Photon Source (Argonne National Laboratory) and processed with XDS (*Kabsch, 2001*). The structure was solved by molecular replacement with Phaser in the Phenix suite using PDB ID 5MF1 (*Fédry et al., 2017*) as search model. The data were originally scaled in R32, but after refinement failed, were scaled in C2. Refinement began with Phenix, and after discovery of three-domain twinning, continued with Refmac of CCP4 (http://www.ccp4.ac.uk/). Intensity based twinning identified fractions of 0.345 with no twin law, 0.338 with twin law −1/2 H-1/2K + L, −1/2 H-1/2 K-L, 1/2 H-1/2K, and 0.317 with twin law −1/2H + 1/2K + L, 1/2 H-1/2K + L, 1/2H + 1/2K. Compared to the previous structure (*Fédry et al., 2017*), ours contains not only more residues, but also two segments with changes in sequence-to-structure. Both segments had mostly residues with small sidechains, making register difficult to determine at 3.3 Å. Register was shifted two positions at residues 281 – 300 as confirmed by the large Phe274 sidechain in a region not built in the 3.3 Å structure. Register was shifted one position at residues 564 – 582 as confirmed by attaching the N-acetylglucosamine residue to Asn578, which is in a Asn-Ala-Thr N-glycosylation sequon, rather than to Thr577, which is not in a known O-glycosylation motif or mucin-like segment. Thr is also not O-linked to N-acetylglucosamine in extracellular proteins except at highly specialized sites in Notch.

## HAP2 trimerization

Purified HAP2 in 20 mM Tris-HCl, pH7.5, 500 mM NaCl (10 mg/ml, 0.1 ml) was incubated with a final concentration of 0.6% n-Dodecyl-β-D-maltoside (DDM) buffer at room temperature for 3 hr and subjected to size exclusion chromatography using a Superdex 200 10/300 GL column in 20 mM Tris-HCl, pH 7.5, 500 mM NaCl, 0.02% DDM. Monomeric material was purified in parallel in absence of DDM. Peak trimer and monomer fractions (*Figure 2—figure supplement 1*) were concentrated to 45 µM for HDX.

## Multi-angle light scattering

Purified HAP2 (250 µg in 0.2 ml) was incubated with 0.1% DDM in 20 mM Tris-HCl, pH7.5, 500 mM NaCl at room temperature for 20 h for trimer preparation. For MALS, untreated monomer or trimer were subjected to gel filtration with a Superdex 200 10/300 GL column (GE Healthcare Life Sciences) in 20 mM Tris-HCl, pH 7.5, 500 mM NaCl with or without 0.1% DDM, respectively using an Agilent liquid chromatography system, a DAWN HELEOS II multi-angle light scattering detector, an Optilab T-rEX refractive index detector, and UV detector (Wyatt Technology Corporation). Data were processed in ASTRA 6 using the protein conjugate model. For monomer, we used d$n$/d$c$ values of 0.185 and 0.145 ml/g for protein and glycan, respectively (*Barer and Joseph, 1954*; *Tumolo et al., 2004*) and A280 extinction value calculated from the HAP2 sequence as 1.166 ml mg$^{-1}$ cm$^{-1}$. We used the weight fraction of protein and glycan of monomer to calculate the d$n$/d$c$ value of the glycoprotein component of the trimer as $\left| \left(\frac{dn}{dc}\right)_{glycoprot} = \left(\frac{dn}{dc}\right)_{prot} * f_{prot} + \left(\frac{dn}{dc}\right)_{glycan} * f_{glycan} \right|$ using $f_{prot}$ and $f_{glycan}$ values of 0.943 and 0.057, respectively. Similarly, the extinction coefficient for the glycoprotein was calculated $\left| \varepsilon_{glycoprot} = \varepsilon_{prot} * f_{prot} + \varepsilon_{glycan} * f_{glycan} \right|$ as 1.099 ml mg$^{-1}$ cm$^{-1}$. For the trimer, we used the derived glycoprotein and published DDM (*Strop and Brunger, 2005*) d$n$/d$c$ values of 0.1827 and 0.133, respectively.

## Hydrogen-deuterium exchange mass spectrometry

Hydrogen deuterium exchange experiments were performed essentially as described (*Iacob et al., 2013*). 45 uM of HAP2 trimer or monomer were diluted 15-fold into 20 mM Tris, 150 mM NaCl, 99% $D_2O$ (pD 8.0) with or without 0.02% n-dodecyl-β-D-maltoside, respectively, at room temperature. At various time points from 10 s to 240 min, an aliquot was quenched by adjusting the pH to 2.5 with an equal volume of 4M Guanidine hydrochloride, 0.2 M potassium phosphate, 0.1 M tris(2-carboxyethyl)phosphine hydrochloride (TCEP-HCl), $H_2O$. Quenched protein was injected into a custom Waters nanoACQUITY UPLC HDX Manager™ [29], digested online using a Poroszyme immobilized pepsin cartridge at 15°C for 30 s, and analyzed on a XEVO G2 mass spectrometer (Waters Corp., USA). The average amount of back-exchange was 18% to 25%, based on analysis of highly deuterated peptide standards. All comparison experiments were done under identical experimental conditions such that deuterium levels were not corrected for back-exchange and are therefore reported as relative (*Wales and Engen, 2006*). Experiments were performed in triplicate independent measurements. The error of measuring the mass of each peptide was ± 0.12 Da. The peptides were identified using PLGS 3.0.1 software and the HDX MS data was processed using DynamX 3.0 (Waters Corp., USA, http://www.waters.com/waters). The common peptides that were compared between the HAP2 monomer and trimer lead to a sequence coverage of 89.9% corresponding to 120 peptic peptides that were followed with hydrogen deuterium exchange uptake plots (*Figure 2—figure supplement 2 and 3-5*).

## Cells and cell culture

*Chlamydomonas reinhardtii* wild type strain *21gr* (mating type *plus*; mt+; CC-1690) and mutant strains *fus1 plus* (*fus1-1*; CC-1158) and *hap2-2 minus* (*40D4*; CC-5281) are available from the *Chlamydomonas* Culture Collection. Cells were grown vegetatively on a rotating shaker in TAP medium on a 13:11 hr light:dark cycle at 22°C (*Gorman and Levine, 1965*). Gametogenesis was induced by transferring vegetatively growing cells into N-free medium in continuous light with aeration as previously described (*Liu et al., 2008*). For some experiments, cells undergoing gametogenesis were cultured on a rotating shaker in continuous light.

## Plasmid construction and transformation into *Chlamydomonas*

Modified versions of the *PsiI* to *NcoI* restriction fragment of *HAP2-HA* plasmid pYJ76 (which also contains a bacterial paramomycin resistance gene, *aphVIII* (*Liu et al., 2015*) were synthesized (Genscript, https://www.genscript.com/) and used to replace the original fragment to generate the following new plasmids: pJJ1, which encodes HAP2-HA modified to remove 2 residues inserted during generation of the original pYJ76 plasmid (HAP2-HA proteins produced by gametes containing pYJ76 and pJJ1 are functionally indistinguishable); pJJ2, which encodes HAP2-HA-W173A/F177A (FH1 for short); pJJ3, which encodes HAP2-HA-F192A/W193A (FH2 for short); and pJJ4, which encodes HAP2-HA-L197A/L200A/I201A (FH3 for short). Codon GCC was used for alanines. All plasmids were confirmed by DNA sequencing. All plasmids encode a 3 x HA tag with spacers (TRGGLSRYPYD VPDYAYPYDVPDYADRSGPYPYDVPDYAASSTRRPPGAS) in HAP2 inserted after residue 702. Plasmids encoding the transgenes were introduced into the *hap2-2* mutant (*40D4*) (*Liu et al., 2015*) by electroporation (*Shimogawara et al., 1998*). DNA samples extracted (Walsh, 1991 #1867860) from colonies of transformants that grew on TAP agar plates containing 10 $\mu$g/ml paramomycin were screened by PCR with primers P18 (5'-CCGATAATGCCTGAACACAATTCCA-3' and P19 (5'- GTATG TCCAGTGGGTCGCTCCAGAAG-3') to detect the transgenes. Expression of HAP2-HA in PCR-positive clones was confirmed by immunoblotting with anti-HA antibody. For simplicity in this manuscript, 40D4 gametes expressing wild type HAP2 tagged with HA are designated *wt*, and the transformants expressing HAP2-FH1, HAP2-FH2, and HAP2-FH3 are designated *fh1, fh2,* and *fh3*, respectively.

## Bioassays for gamete functions during fertilization

### Ciliary adhesion

The ability of gametes to undergo ciliary adhesion was quantified with an electronic particle counter (Beckman Z2 Coulter Counter) (*Snell and Roseman, 1979*). Briefly, 5 min after equal numbers (2 × $10^6$ cells/ml in N-free medium) of *plus* and *minus* gametes were mixed together, the number of cells

that had adhered was determined by measuring the number of single cells that had been lost from the samples.

## Gamete signaling/activation

The ability of gametes to lose their cell walls upon mixing, which is a measure of ciliary adhesion-induced signaling and gamete activation, was determined as described (Snell, 1982). Briefly, plus and minus gametes ($5 \times 10^7$ cells/ml in N-free medium) were mixed for 10 min, added to 1.6 volumes of ice cold N-free medium containing 0.075% Triton-X 100 and 5 mM EDTA, briefly vortexed, subjected to centrifugation (8700 x g for 30 s), and the $OD_{435}$ of the supernatant was determined immediately using a Nanodrop 2000 spectrophotometer (Thermo Scientific). The $OD_{435}$ of a sample of similarly treated gametes that had first been disrupted by sonication (3 times for 5 s each on ice (Microson XL sonicator) was used as a measure of 100% cell wall loss.

## Mating structure adhesion

The ability of gametes to adhere by their mating structures was quantified by phase contrast microscopy. 10 min after equal numbers of plus and minus gametes ($2 \times 10^7$ cells/ml in N-free medium) were mixed together, cells were fixed with an equal volume of 5% glutaraldehyde, ciliary adhesions were disrupted by pipetting 10 times with a 1 ml pipette tip, and the percent of cells present as pairs was determined by microscopy. fus1 plus mutant gametes, which lack the FUS1 adhesion protein on their mating structure, fail to adhere by their mating structures and served as a negative control. Fusion-defective minus mutant hap2 gametes, which undergo normal mating structure adhesion, served as a positive control. At least 200 cells were counted for each sample.

## Gamete fusion

The ability of gametes to fuse to form zygotes, which have four cilia as opposed to unfused gametes, which have two cilia, was quantified by phase contrast microscopy. Plus and minus gametes in equal numbers were mixed for 5 – 30 min, fixed with an equal volume of 5% glutaraldehyde, and the percent of single cells that had fused to form quadri-ciliated zygotes was determined (2 x number of quadri-ciliated cells/([2 x number of quadri-ciliated cells + number of single gametes] x 100) (Liu et al., 2008). At least 200 cells were counted for each sample.

## SDS-PAGE and immunoblotting

Gametes ($5 \times 10^6$ in N-free medium) were harvested by centrifugation, resuspended in 25 microliters of 20 mM HEPES pH 7.2, 5 mM $MgCl_2$, 1 mM DTT, 1 mM EDTA, 25 mM KCl containing a protease inhibitor cocktail (Roche Applied Science), mixed with an equal volume of 2x SDS-PAGE sample buffer (80 mM Tris-HCl, pH 6.8, 2 % SDS, 10% glycerol, 0.0006% Bromophenol blue, 20 mM TCEP, 2 mM EDTA), incubated at 98 °C for 5 min, clarified by a brief centrifugation, and one half ($2.5 \times 10^6$ gametes) was subjected to SDS 7% PAGE followed by immunoblotting with HA antibody (Misamore et al., 2003). To assess surface localization of HAP2-HA, live gametes were treated with 0.05% trypsin before SDS-PAGE (Liu et al., 2010).

## Indirect immunofluorescence

Indirect immunofluorescence was carried out as described previously (Liu et al., 2008), with the following minor modifications. Briefly, after fixing gametes in ice-cold methanol for 20 min, samples were blocked with goat serum and probed with 1/100 rat anti-HA (Sigma) to detect HA-tagged HAP2. Samples were then washed with 1x PBS, stained with 1:400 Alexa Fluor 488-conjugated goat anti-rat IgG (Invitrogen), and mounted using Fluoromount-G prior to visualization on a Leica SP5 X confocal microscope.

## Acknowledgements

Supported by NIH grants R01-CA31798 to TAS, R01-GM56778 and R35-GM122565 to WJS, and a research collaboration with the Waters Corporation (JRE). We thank Prof. Thomas Wales for helpful discussions on the HDX experiments and Margaret Nielsen for illustrations.

## Additional information

### Funding

| Funder | Grant reference number | Author |
|---|---|---|
| National Institutes of Health | R01-CA31798 | Timothy A Springer |
| National Institutes of Health | R01-GM56778 | William J Snell |
| National Institutes of Health | R35-GM122565 | William J Snell |
| Waters Corporation | Research Collaboration | John R Engen |

The funders had no role in study design, data collection and interpretation, or the decision to submit the work for publication.

### Author contributions

Juan Feng, Investigation, Methodology, Writing—original draft, Writing—review and editing, Designed the experiments, Carried out the biochemical studies and crystallization, Solved and refined the structure, Wrote the manuscript; Xianchi Dong, Investigation, Methodology, Writing—original draft, Writing—review and editing, Designed the experiments, Solved and refined the structure, Wrote the manuscript; Jennifer Pinello, Jun Zhang, Investigation, Methodology, Writing—original draft, Writing—review and editing, Designed the experiments, Performed experiments with Chlamydomonas cells, Wrote the manuscript; Chafen Lu, Methodology, Writing—original draft, Writing—review and editing, Designed the experiments, Wrote the manuscript; Roxana E Iacob, Investigation, Methodology, Writing—review and editing, Designed the experiments, Performed HDX, Wrote the manuscript; John R Engen, William J Snell, Investigation, Methodology, Writing—review and editing, Designed the experiments, Wrote the manuscript; Timothy A Springer, Supervision, Investigation, Methodology, Writing—original draft, Writing—review and editing, Designed the experiments, Wrote the manuscript

### Author ORCIDs

Timothy A Springer (iD) http://orcid.org/0000-0001-6627-2904

### Decision letter and Author response

Decision letter https://doi.org/10.7554/eLife.39772.021
Author response https://doi.org/10.7554/eLife.39772.022

## Additional files

### Supplementary files

• Supplementary file 1. Crystal data collection and refinement statistics. (a) The numbers in parentheses refer to the highest resolution shell. (b) Rmerge = $\Sigma h \, \Sigma i \, |Ii(h) - <I(h)>| / \Sigma h \Sigma i \, Ii(h)$, where Ii(h) and <I(h)> are the i$^{th}$ and mean measurement of the intensity of reflection h. (c) Pearson's correlation coefficient between average intensities of random half-datasets for unique reflection (*Karplus and Diederichs, 2012*). (d) Rfactor = $\Sigma h ||Fobs(h)| - |Fcalc(h)|| / \Sigma h |Fobs(h)|$, where Fobs (h) and F calc (h) are the observed and calculated structure factors, respectively. No I/σ(I) cutoff was applied. e Calculated with MolProbity (*Davis et al., 2007*).
DOI: https://doi.org/10.7554/eLife.39772.016

• Transparent reporting form
DOI: https://doi.org/10.7554/eLife.39772.017

### Data availability

Diffraction data has been deposited in PDB under the accession code 6DBS. All data generated or analysed during this study are included in the manuscript and supporting files.

The following dataset was generated:

| | | **Database, license,** |
|---|---|---|

| Author(s) | Year | Dataset title | Dataset URL | and accessibility information |
|---|---|---|---|---|
| Feng J, Dong X, Pinello JF, Zhang J, Lu C, Iacob RE, Engen JR, Snell WJ, Springer TA | 2018 | Data from: Fusion surface structure, function, and dynamics of gamete fusogen HAP2 | https://www.rcsb.org/structure/6DBS | Publicly available at the RCSB Protein Data Bank (accession no: 6DBS) |

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
