## [Decision Letter]

Thank you for submitting your article "Fusion surface structure, function, and dynamics of gamete fusogen HAP2" for consideration by *eLife*. Your article has been reviewed by two peer reviewers, and the evaluation has been overseen by a Reviewing Editor and Andrea Musacchio as the Senior Editor. The reviewers have opted to remain anonymous.

The reviewers have discussed the reviews with one another and the Reviewing Editor has drafted this decision to help you prepare a revised submission.

Summary:

This study extends recent work by the Rey lab on the membrane fusogen HAP2 that mediates gamete fusion and that resembles viral class II fusion proteins such as those found on viruses like *dengue* and *West Nile*. The new structure reported here reveals that the fusion loops are not organized as those of class II viral fusion glycoproteins and that they form three helices with apically-pointing hydrophobic residues. The authors mutated those residues and demonstrated that the hydrophobic residues in the axis-proximal α1- and α2-helices are more important for fusion activity than those in the axis-distal η1 helix. The apex of domain D2 (D2.2) has not the same orientation in the two crystal structures suggesting that D2.2 can tilt at its junction with D2.1, and that the fusions loop can rock at Arg185. Besides the structure, the authors also characterized HAP2 dynamics by HDX in both its monomeric and trimeric states revealing D2 flexibility.

Essential revisions:

Both reviewers and the reviewing editor appreciated the quality of the work and concluded that it adds substantial information beyond that published by the Rey laboratory. There were no substantial concerns, and thus only minor revisions are required as detailed below:

*Reviewer #1:*

The authors consider that they have crystallized an intermediate. This may be correct. However, the "hotspot" argument is not convincing. The "hotspot" model applies to protein-protein interactions and probably not to protein membrane interaction.

Introduction, last paragraph: the authors have not characterized an intermediate on the "fusion pathway" (there is no membrane in the crystal) but rather an intermediate on the "structural transition pathway".

Subsection “HAP2 dynamics and flexibility within D2”, last paragraph: about the four polypeptide segments forming the D2.1-D2.2 junction, the authors write that none of them contain secondary structure. This is not correct; in fact, none of them contain canonical secondary structure.

Figure 4B is not easy to read/decipher…

Figures 4C and 2C are redundant.

In Figure 7E, F and G, the meaning of p is not clear. Particularly in Figure F: I would like to know with which samples p is associated

Reviewer #2:

My suggestions are to make the paper more accessible to a non-structural biologist! I was able to figure out the hydrogen deuterium exchange studies, but only by reading some other material, and I have no idea what Dali searches are and was too lazy to look this up. I suspect the authors could try to put themselves of the shoes of a virologist and do some nice editing so that we non-structural souls can follow the experiments more easily. Nice job in describing the structure and how it has both similarities and differences to previous studies, including other class II fusion proteins.

---

## [Author Response]

Reviewer #1:

The authors consider that they have crystallized an intermediate. This may be correct. However, the "hotspot" argument is not convincing. The "hotspot" model applies to protein-protein interactions and probably not to protein membrane interaction.

While the hotspot model has only been demonstrated for protein-protein interactions, the reviewer proposes no reason why it should not be applicable to protein-membrane interactions. The physical chemistry of protein-protein interactions holds for both water-soluble proteins and membrane embedded proteins. Hydrophobic and van der Waals interactions drive protein-protein interactions for water-soluble proteins. Hydrophobic exposure to solvent is minimized. Membrane-embedded proteins interact by minimizing exposure of polar surface to the lipid environment. The importance of residues in the middle of protein-protein interfaces is seen in both types of interactions. Lipids have both hydrophobic and polar regions like proteins. Thus we believe that the hotspot model will also hold for protein-lipid interactions.

Introduction, last paragraph: the authors have not characterized an intermediate on the "fusion pathway" (there is no membrane in the crystal) but rather an intermediate on the "structural transition pathway".

We agree, and have so revised.

Subsection “HAP2 dynamics and flexibility within D2”, last paragraph: about the four polypeptide segments forming the D2.1-D2.2 junction, the authors write that none of them contain secondary structure. This is not correct; in fact, none of them contain canonical secondary structure.

Thank you, so corrected.

Figure 4B is not easy to read/decipher.

There were unnecessary residue labels that have been removed to make the panel easier to appreciate.

Figures 4C and 2C are redundant.

We partially agree but prefer to keep both. Figure 2C is color-coded according to HDX and shows that Arg185 and its loop exchange much more rapidly than the carbonyl cage. Only Arg185 is labeled, making it easier for the reader to focus on the key idea of the carbonyl cage without interference from unneeded labeling. Figure 4C has a simpler color scheme making it much easier to see the hydrogen bonded O and N atoms and follow the story of Figure 4 in the same color scheme. And it labels the residues that hydrogen bond to Arg185 and shows that most that contribute to the cage are in in PFAM 10699, thereby providing a reasonable explanation for conservation of the “HAP2/GCS1 domain”. Actually, Figure 2C and 4C are each very small and are placed within their figures in space that will just be wasted white space if we remove one of them. So there is very little gain to removing one of them.

In Figure 7E, F and G, the meaning of p is not clear. Particularly in Figure F: I would like to know with which samples p is associated

We thank the reviewer for pointing out the need for clarification. We provide a revised Figure 7. The experiments in these 3 parts of Figure 7 present results from bioassays showing that the fusion helices mutants fh1, fh2, and fh3 are competent in all of the steps leading up to membrane fusion, but, like the parental hap2 mutants, are blocked at bilayer merger. In the legend, we have now indicated the statistical tests which led to the p-values shown in the figure. Graphpad PRISM software was used to perform all statistical analyses for the assays in Figure 7 and is listed in our key resource table. For each assay, the following statistical analysis was performed: first, a non-parametric ANOVA (the Kruskal Wallis test) was applied to identify whether a significant difference existed among the multiple groups assayed for each variable (ciliary adhesion, gamete activation, and mating structure adhesion) tested. Second, if a statistically significant difference was observed among the sample groups for a particular variable, Dunn’s post test (a follow-up test to the Kruskal-Wallis) was then performed to identify the pairwise comparisons between sample groups that were significantly different from one another. The lack of clarity noted by Reviewer #1 in (F) likely arose because we reported the insignificant p-value for the Kruskal Wallis test (all 6 samples were indistinguishable in extent of gamete activation, and a Dunn’s post-test for pairwise comparisons was unnecessary). To reduce potential reader confusion, we have now decided to limit the number of p-values displayed in (E-G). We have removed the expected Kruskal-Wallis test’s insignificant p-value in (F) as well as the p-value in (E) documenting the expected reduced ciliary adhesion in the wt- x wt+. These expected results are now described in the legend. We retained the insignificant pairwise p-values in (G). We believe these simplifications now highlight the most important result from the assays, namely, that the fusion helix mutants (fh1-, fh2-, and fh3-) are indistinguishable from their parental hap2 mutants in ciliary adhesion, gamete activation, and mating structure adhesion. Thus, their inability to form zygotes is specifically a consequence of their inability to bring about bilayer merger.

Reviewer #2:

My suggestions are to make the paper more accessible to a non-structural biologist! I was able to figure out the hydrogen deuterium exchange studies, but only by reading some other material, and I have no idea what Dali searches are and was too lazy to look this up. I suspect the authors could try to put themselves of the shoes of a virologist and do some nice editing so that we non-structural souls can follow the experiments more easily. Nice job in describing the structure and how it has both similarities and differences to previous studies, including other class II fusion proteins.

We replaced “Dali searches” with “Search for structurally homologous domains” and added a reference.